# GEOMETRIC KOLMOGOROV SUPERPOSITION REPRESENTATION OF GROUP INVARIANT FUNCTION FOR COMPUTATIONAL SCIENCE

## ABSTRACT

The Kolmogorov-Arnold Theorem (KAT), or more generally, the Kolmogorov Superposition Theorem (KST), establishes that any non-linear multivariate function can be exactly represented as a finite superposition of non-linear univariate functions. Unlike the universal approximation theorem, which provides only an approximate representation without guaranteeing a fixed network size, KST offers a theoretically exact decomposition. The Kolmogorov-Arnold Network (KAN) was introduced as a trainable model to implement KAT, and recent advancements have adapted KAN using concepts from modern neural networks. However, KAN struggles to effectively model physical systems that require inherent equivariance or invariance geometric symmetries as $E(3)$ transformations, a key property for many scientific and engineering applications. In this work, we propose the Geometric Kolmogorov Superposition Representation (GKSR), a novel extension of KAT, and Geometric Kolmogorov Superposition Network (GKSN), its implementation, which incorporate invariance over various group actions, including $O(n)$, $O(1, n)$, $S_n$ and general $GL$, enabling accurate and efficient modeling of these systems. Our approach provides a unified approach that bridges the gap between mathematical theory and practical architectures for physical systems, expanding the applicability of KAN to a broader class of problems. We provide experimental validation on molecular dynamical systems and particle physics. [1]

## 1 INTRODUCTION

Kolmogorov Arnold Networks (KANs) (Liu et al., 2025) have recently risen to the interest of the machine learning community as an alternative to the well-consolidated Multi-Layer Perceptrons (MLPs) (Hornik et al., 1989). KAN are based on the Kolmogorov-Arnold Theorem (KAT, (Kolmogorov, 2009)). KAT was developed to solve Hilbert's 13th problem and describes how to exactly and with a finite and known number of univariate functions represent any multivariate continuous function. KAT has found multiple applications in mathematics (Laczkovich, 2021), fuzzy logic (Kreinovich et al., 1996), pattern recognition (Köppen, 2002), and neural networks (Kůrková, 1992; Liu et al., 2025).

Multiple extensions of KAN have been proposed (Ji et al., 2024), either as a plug-in replacement of MLP (Xu et al., 2024b; Carlo et al., 2024), or as a surrogate function (Abueidda et al., 2024; Wang et al., 2024; Shuai & Li, 2024). KANs have been also extended by exploring alternative basis functions such as Chebychev polynomials (Sidharth et al., 2024; Mostajeran & Faroughi, 2024), wavelet functions (Bozorgasl & Chen, 2024), Fourier series (Xu et al., 2024a), or alternative representations (Guilhoto & Perdikaris, 2024).

Function representation in scientific computing requires modeling physical symmetries (Finzi et al., 2021; Goodman & Wallach, 2009; Noether, 1971). Interatomic potentials used in chemistry are invariant to translations, rotations, and reflections (i.e. $E(3)$ group). The need to model symmetries can also be found in fluid dynamics, astrophysics, material science, and biology. While MLP-based architectures have been widely explored (Schütt et al., 2017; Batatia et al., 2023; Satorras et al., 2022; Liao & Smidt, 2023; Zaverkin et al., 2024), it is not clear how to model a physical system with

---

[1] For the code, see Appendix K and `https://anonymous.4open.science/r/GKSN-37BD/`.

KAN-based architectures, especially since KAN models have shown potential to overcome the curse of dimensionality (Lai & Shen, 2021; Poggio, 2022).

Our contributions are : i) to propose a group invariant representation to $O(n)$, $O(1, n)$ and in general $GL(n)$ symmetries (Section 4). We further extend the results to include the permutation invariance with respect to input data, which reduces the parameter count of the network and improves generalization. ii) After providing the theoretical justification, we present practical architectures (Section 5) and investigate their performances with scientifically inspired experiments. We investigate the learning capability of the proposed KAN model for an idealized model (Section 6.2), which allows us to simulate multiple particles in multiple dimensions. iii) To further investigate the learning capability of the proposed model, we experiment on real datasets for material design, the MD17 (Section 6.2) and MD22 (Section 6.2); but also particle physics with Top-tagging (Section 6.2) and Quark-gluon tagging (Section 6.2). iv) Extensive formal theorems and proofs are provided in the supplementary material (Section A) to support our claims summarized in Table 1.

## 2 RELATED WORKS

**Symmetry preserving machine learning architecture**   Machine learning interatomic potentials (MLIPs) have emerged as powerful tools for modeling interatomic interactions in molecular and materials systems, offering a computationally efficient alternative to traditional ab initio methods. Architectures like Schnet (Schütt et al., 2017) use continuous-filter convolutional layers to capture local atomic environments and message passing, enabling accurate predictions of molecular properties. To further enhance physical expressivity, $E(3)$-equivariant architectures (Thomas et al., 2018b) have been developed, which respect the symmetries of Euclidean space (rotations, translations, and reflections) by design. These models ensure that predictions of energies and forces are invariant, respectively equivariant, to group actions of $E(3)$, making them highly data-efficient for tasks like force field prediction in molecular dynamics. Equivariant or invariant architectures enhance data efficiency, accuracy, and physical consistency in tasks where input symmetries (e.g., rotation, reflection, translation) dictate output invariance or equivariance. Symmetry-preserving architectures for the Lorentz group have been proposed, based on high-order tensor products as LoLa (Butter et al., 2018), LBN (Erdmann et al., 2019) LGN (Bogatskiy et al., 2020), and LorentzNet (Gong et al., 2022), which introduce Minkowski dot product attention. Finally, permutation preserving models have been proposed to model functions over sets, as DeepSet and subsequent models (Zaheer et al., 2017; Amir et al., 2023). The advantage of KAN architecture has not yet been explored; we thus take a fundamental step in this direction with our study.

**Kolmogorov-Arnold Network Architecture**   Kolmogorov-Arnold Networks (KANs) apply univariate function representation in a multi-layer system and propose to use splines as the basis functions to approximate the univariate functions. Early work by Hecht-Nielsen (1987) introduced one of the first neural network architectures based on Kolmogorov–Arnold representation theorem, demonstrating its potential capability for efficient function approximation. Lai & Shen (2021) studies the approximation capability of KAT-based models in high dimensions and how they could potentially break the curse of dimensionality (Poggio, 2022). Ferdaus et al. (2024) propose to combine Convolutional Neural Networks (CNNs) with KAN architecture. Furthermore, Yang & Wang (2025) explored the integration of KAN principles into transformer models, achieving improvements in efficiency for sequence modeling tasks. Hu et al. (2025) propose EKAN, an approximation method for incorporating matrix group equivariance into KANs. While these studies highlight the versatility of KAN architectures in adapting to various neural network architectures, the extension to physical and geometrical symmetries has not been fully considered.

**Theoretical Work on KAN**   KANs are rooted in the Kolmogorov–Arnold representation theorem, established by Andrey Kolmogorov (Kolmogorov, 1957) and later refined by Vladimir Arnold (Arnold, 1959). Building upon this foundation, David Sprecher (Sprecher, 1965) and George Lorentz (Lorentz, 1976) provided constructive algorithms to implement the theorem, enhancing its applicability in computational contexts. Recent theoretical advancements have addressed challenges in training KANs, such as non-smooth optimization landscapes. Researchers have proposed various techniques to improve the stability and convergence of KAN training, including regularization methods like dropout and weight decay (Braun & Griebel, 2009), as well as optimization strategies involving

adaptive learning rates, while Igelnik & Parikh (2003) have proposed using cubic spline as activation and internal function for efficient approximation. These contributions have reduced the gap between the mathematical foundations of KANs and their practical implementation in machine learning. However, training with energies requires including physics symmetries. In this work, we demonstrate how extending the KAN architecture enhances the learning capacity of KAT-based models.

## 3 BACKGROUND

**Equivariance and invariance** We call a function $\phi : X \to Y$ *equivariant* or *invariant*, if given two families of multiplicative transformations on the input space $X$, $\{T_g^X \, ; \, T_g^X : X \to X, \, g \in G\}$, and on the output space $Y$, $\{T_g^Y \, ; \, T_g^Y : Y \to Y, \, g \in G\}$, the following relations hold

$$\underbrace{\phi(T_g^X(\boldsymbol{x})) = T_g^Y(\phi(\boldsymbol{x}))}_{\text{equivariant}} \quad \text{or} \quad \underbrace{\phi(T_g^X(\boldsymbol{x})) = \phi(\boldsymbol{x})}_{\text{invariant}}, \quad \forall x \in X, \, \forall g \in G. \tag{1}$$

An example of $\phi$ is a non-linear transformation that maps a multivariate variable $\boldsymbol{x} = (\boldsymbol{x}_1, \ldots, \boldsymbol{x}_m) \in \mathbb{R}^{m \times n}$ representing a point cloud with $m$ points to a point $\boldsymbol{y} = \phi(\boldsymbol{x}) \in \mathbb{R}^{m \times n}$, with $T_g$ a translation of the input $T_g^X(\boldsymbol{x}) = \boldsymbol{x} + \boldsymbol{g}$ and an associated translation $T_g^Y(\boldsymbol{y}) = \boldsymbol{y} + \boldsymbol{g}$ in the output domain for $g \in G = \mathbb{R}^{m \times n}$. When $\phi$ is equivariant with respect to the action of $G$, then first applying $T_g^X$ and then applying $\phi$ coincides with first applying $\phi$ and then $T_g^Y$. When $\phi$ is invariant with respect to $G$, then applying the translation does not alter the output, i.e., $\phi(\boldsymbol{x} + \boldsymbol{g}) = \phi(\boldsymbol{x})$. In this work, we consider the following three types of symmetries:

- *translation symmetry*: $\phi(\boldsymbol{x} + \boldsymbol{g}) = \phi(\boldsymbol{x})$ for the invariance and $\phi(\boldsymbol{x} + \boldsymbol{g}) = \phi(\boldsymbol{x}) + \boldsymbol{g}$ for equivariance, with $\boldsymbol{g} \in \mathbb{R}^n$ and where $\boldsymbol{x} + \boldsymbol{g}$ refers to the element-wise operation $(\boldsymbol{x}_1 + \boldsymbol{g}, \ldots, \boldsymbol{x}_m + \boldsymbol{g})$;
- *rotation and reflection symmetry*: given an orthogonal matrix $\boldsymbol{Q} \in \mathbb{R}^{n \times n}$, $\phi$ is invariant or equivariant if $\phi(\boldsymbol{Qx}) = \phi(\boldsymbol{x})$ or $\phi(\boldsymbol{Qx}) = \boldsymbol{Q}\phi(\boldsymbol{x})$, and where $\boldsymbol{Qx}$ refers to the element-wise operation $(\boldsymbol{Qx}_1, \ldots, \boldsymbol{Qx}_m)$;
- *permutation symmetry*: $\phi$ is invariant or equivariant, if $\phi(\boldsymbol{x}_1, \ldots, \boldsymbol{x}_m) = \phi(\boldsymbol{x}_{\pi_1}, \ldots, \boldsymbol{x}_{\pi_m})$ and $\phi(\pi(\boldsymbol{x})) = \pi(\phi(\boldsymbol{x}))$, for any permutation $\pi : [m] \to [m]$.

**First Fundamental Theorem of** $GL(V)$ According to the First Fundamental Theorem (FFT) (Kraft & Procesi, 1996), the ring of invariant polynomial functions can be generated by the invariants of the symmetry group. Villar et al. (2021) shows how the FFT can be used to represent, among others, $O(n)$ and $O(1, n)$ invariant functions and their use in MLP. The FFT states that the ring of invariants for the action of $\mathrm{GL}(V)$ on $V^p \oplus V^{*q}$ is generated by the invariants $(i \mid j)$: $K[V^p \oplus V^{*q}]^{\mathrm{GL}(V)} = K[(i \mid j) \mid i = 1, \ldots, p, \, j = 1, \ldots, q]$. Appendix A contains additional information.

**Kolmogorov-Arnold representation theorem** KAT provides a powerful theoretical tool to represent a multivariate function $f(x_1, \ldots, x_m)$ as the composition of functions of a single variable. The original form of KAT states that a given continuous function $f : [0, 1]^m \to \mathbb{R}$ can be represented as

$$f(x_1, \ldots, x_m) = \sum_{q=1}^{2m+1} \psi_q \big( \sum_{p=1}^{m} \phi_{qp}(x_p) \big) \tag{2}$$

with $\psi_q : \mathbb{R} \to \mathbb{R}$ and $\phi_{qp} : [0, 1] \to \mathbb{R}$ uni-variate continuous functions. Kolmogorov superposition theorems (KST) refer to extensions of the original KAT (Sprecher, 1965).

**Ostrand superposition theorem (OST)** Ostrand (1965) proposed an extension of the original KAT to input compact domains. The theorem states that, given compact metric spaces $\{X^p\}_{i=1}^m$ of finite dimension $d_p = |X^p|$, such that $\sum_{p=1}^m d_p = M$, a continuous function $f : \prod_{p=1}^m X^p \to \mathbb{R}$ is representable in the form

$$f(\boldsymbol{x}_1, \ldots, \boldsymbol{x}_m) = \sum_{q=1}^{2M+1} \psi_q \left( \sum_{p=1}^{m} \phi_{qp}(\boldsymbol{x}_p) \right) \tag{3}$$

Table 1: Kolmogorov superposition formulas (Guilhoto & Perdikaris, 2024) for a continuous function $f(x_1, \ldots, x_d)$ or $f(\boldsymbol{x}_1, \ldots, \boldsymbol{x}_m)$ and their complexity in terms of parameters. $\langle \boldsymbol{x}_i, \boldsymbol{y}_j \rangle$ is either Euclidean or Minkowski scalar product, while $(\boldsymbol{x}_i | \boldsymbol{y}_j)$ is the $GL(n)$ invariant as defined in Section A.

| Version | Formula | Inner Functions | Outer Functions | Other Parameters | Symmetry Group |
|---|---|---|---|---|---|
| Kolmogorov (1957) | $\sum_{q=1}^{2m+1} \psi_q \left( \sum_{p=1}^{m} \phi_{q,p}(x_p) \right)$ | $(2m+1)m$ | $2m+1$ | N/A | - |
| Ostrand (1965) | $\sum_{q=1}^{2mn+1} \psi_q \left( \sum_{p=1}^{d} \phi_{q,p}(\boldsymbol{x}_p) \right)$ | $(2nm+1)m$ | $2mn+1$ | N/A | - |
| Lorentz (1962) | $\sum_{q=1}^{2m+1} \psi \left( \sum_{p=1}^{m} \lambda_p \phi_q(x_p) \right)$ | $2m+1$ | $1$ | $\lambda \in \mathbb{R}^m$ | - |
| Sprecher (1965) | $\sum_{q=1}^{2m+1} \psi_q \left( \sum_{p=1}^{m} \lambda_p \phi(x_p + qa) \right)$ | $1$ | $2m+1$ | $a \in \mathbb{R}, \lambda \in \mathbb{R}^d$ | - |
| Kurkova (1991) | $\sum_{q=1}^{N} \psi \left( \sum_{p=1}^{m} w_{pq} \phi_q(x_p) \right)$ | $2m+1 \le N$ | $1$ | $w \in \mathbb{R}^{m \times N}$ | - |
| Laczkovich (2021) | $\sum_{q=1}^{N} \psi \left( \sum_{p=1}^{d} \lambda_{pq} \phi_q(x_p) \right)$ | $N$ | $1$ | $\lambda \in \mathbb{R}^{m \times N}$ | - |
| **This work** Theorem A.11 | $\sum_{q=1}^{2m^2+1} \psi_q \left( \sum_{i=1,j=1}^{m,m} \phi_{qij}(\langle \boldsymbol{x}_i, \boldsymbol{x}_j \rangle) \right)$ | $(2m^2+1)m^2$ | $2m^2+1$ | N/A | $O(n), O(1,n)$ |
| **This work** Theorem A.12 | $\sum_{q=1}^{2mn+1} \psi_q \left( \sum_{i=1,j=1}^{m,n} \phi_{qij}(\langle \boldsymbol{x}_i, \boldsymbol{y}_j \rangle) + \sum_{i=1,j=1}^{n,n} \phi'_{qij}(\langle \boldsymbol{y}_i, \boldsymbol{y}_j \rangle) \right)$ | $(2mn+1) \times (mn+n^2)$ | $2mn+1$ | N/A | $O(n), O(1,n)$ |
| **This work** Theorem A.13 | $\sum_{q=1}^{2mn+1} \psi_q \left( \sum_{i=1,j=1}^{m,n} \phi_{qij}(\langle \boldsymbol{x}_i, \boldsymbol{x}_j \rangle) \right)$ | $(2mn+1)mn$ | $2mn+1$ | N/A | $O(n), O(1,n)$ |
| **This work** Theorem A.10 | $F \left( \sum_{i=1}^{m} \phi_1(x_i), \ldots, \sum_{i=1}^{m} \phi_{2m+1}(x_i) \right)$ | $M = (2m+1)$ | $(2M+1)M$ | N/A | $S_n$ |
| **This work** Theorem A.5 | $\sum_{q=1}^{2m^2+1} \psi_q \left( \sum_{i=1,j=1}^{m,m} \phi_{qij}((\boldsymbol{x}_i | \boldsymbol{x}_j)) \right)$ | $(2m^2+1)m^2$ | $2m^2+1$ | N/A | $GL(n)$ over polynomial ring |

with $\boldsymbol{x}_p \in X^p$, and $\phi_{qp} : X^p \to \mathbb{R}$ continuous functions. When $d_p = n$ for all $p$, then $M = nm$. The difference between KAT and OST is, that the building functions $\phi_{qp}$ in OST are not on scalars (not any more uni-variate), but defined over arbitrary compact spaces $X^p$ (thus multivariate).

Although the original formulation has been criticized (Girosi & Poggio, 1989), other versions of KAT have been proposed as counter-arguments to the smoothness and efficiency of the representation (Kůrková, 1991). Table 1 summarizes the various versions of the KAT (Kolmogorov, 1957; Braun, 2009; Kůrková, 1991; Kůrková, 1992; Laczkovich, 2021; Sprecher, 1965; 1996).

## 4 GEOMETRIC KOLMOGOROV SUPERPOSITION REPRESENTATION (GKSR)

We want to extend the KST to invariant functions to the action of $O(n)$ or $O(1,n)$. While the original KST already tells us that we can represent the original function as the superposition of univariate functions (see Equation 2), which requires a total of $(mn+1)(2mn+1)$ univariate functions, we would like to have a better form of this representation. By the OST, any $F : (\mathbb{R}^n)^m \to \mathbb{R}$ can be represented using only $(m+1)(2mn+1)$ auxiliary functions, each mapping $\mathbb{R}^n \to \mathbb{R}$; hence they are $n$-variate rather than univariate. However, we claim that we can represent a generic invariant function $f(\boldsymbol{x})$ using only univariate functions, as

$$f(\boldsymbol{x}_1, \ldots, \boldsymbol{x}_m) = \sum_{q=1}^{2m^2+1} \psi_q \left( \sum_{i=1,j=1}^{m,m} \phi_{qij}(\langle \boldsymbol{x}_i, \boldsymbol{x}_j \rangle) \right), \tag{4}$$

more formally stated and proved in Theorem A.11. The result is based on the fact that the set $\langle \boldsymbol{x}_i, \boldsymbol{x}_j \rangle_{i,j=1}^{n}$, where $\langle \boldsymbol{x}_i, \boldsymbol{x}_j \rangle$ is either the Euclidean or Minkowski scalar product, represents a complete set of invariant features (Villar et al., 2021). Unfortunately, this form is $m^4$ in the number of nodes. In Theorem A.12, we provided an improved version of the geometric KST that grows $m^2$ with the number of nodes, since it only uses a linear number of invariant features. Indeed, if we select $\boldsymbol{y}_j = \alpha_j(\boldsymbol{x}_1, \ldots, \boldsymbol{x}_m)$ a linear combination of the inputs such that they span the full space $\mathbb{R}^n$:

$$f(\boldsymbol{x}_1, \ldots, \boldsymbol{x}_m) = \sum_{q=1}^{2mn+1} \psi_q \left( \sum_{\substack{1 \le i \le m, \\ 1 \le j \le n}} \phi_{qij}(\langle \boldsymbol{x}_i, \boldsymbol{y}_j \rangle) + \sum_{\substack{1 \le i \le n, \\ 1 \le j \le n}} \phi'_{qij}(\langle \boldsymbol{y}_i, \boldsymbol{y}_j \rangle) \right).$$

While the formal statement and proof are given in Theorem A.12, the intuition is that we can project the input on the vectors $\boldsymbol{y}_j$. Since these vectors, built as linear combinations of the input, do not

Figure 1: The geometric Kolmogorov superposition network is composed of layers that comprise two terms. The first term is based on the classical KST function representation, while the second term, similar to a residual path, is an almost linear term that helps the training of the non-linear functions.

form an orthonormal basis, we need the information of their inner product $\langle \boldsymbol{y}_i, \boldsymbol{y}_j \rangle$ to reconstruct the invariant features $\langle \boldsymbol{x}_i, \boldsymbol{x}_j \rangle$. If we further restrict the vectors $\boldsymbol{y}_j$ to be a fixed subset of the input features, we have (see Theorem A.13),

$$f(\boldsymbol{x}_1, \ldots, \boldsymbol{x}_m) = \sum_{q=1}^{2mn+1} \psi_q \left( \sum_{i=1, j=1}^{m,n} \phi_{qij}(\langle \boldsymbol{x}_i, \boldsymbol{x}_j \rangle) \right), \tag{5}$$

which further reduces the need for the additional $n^2$ invariant features. We now formalize the preceding result; a more detailed theoretical derivation is provided in Appendix A:

---
**$O(n)$ invariance - v3**

**Corollary 4.1.** *Suppose that* $\mathrm{span}(\{\boldsymbol{x}_j\}_{j=1}^n) = \mathbb{R}^n$. *Then, a continuous function invariant to the action of* $O(n)$ $f(\boldsymbol{x}_1, \ldots, \boldsymbol{x}_m) : X^m \to \mathbb{R}$, *with* $X \subset \mathbb{R}^n$ *a compact space, can be represented as* $f(\boldsymbol{x}_1, \ldots, \boldsymbol{x}_m) = \sum_{q=1}^{2mn+1} \psi_q \left( \sum_{i=1, j=1}^{m,n} \phi_{qij}(\langle \boldsymbol{x}_i, \boldsymbol{x}_j \rangle) \right).$
---

**Equivariant $O(n)$ or $O(1,n)$ functions** In the supplementary material (Appendix A.6), we discuss the equivariant versions, which can be built from invariant functions (Villar et al., 2021), as

$$f^{\mathrm{lin}}(\boldsymbol{x}_1, \ldots, \boldsymbol{x}_m) = \sum_{l=1}^{m} f_l(\boldsymbol{x}_1, \ldots, \boldsymbol{x}_m)\boldsymbol{x}_l, \quad f^{\mathrm{grad}}(\boldsymbol{x}_1, \ldots, \boldsymbol{x}_m) = \sum_{l=1}^{m} \nabla_{\boldsymbol{x}_l} f_l(\boldsymbol{x}_1, \ldots, \boldsymbol{x}_m),$$

where $f^{\mathrm{lin}} \in C^1$ and $f_l$ is an invariant $C^1$-function.

**Translation and permutation symmetry** Translation invariance is achieved by centering the input, i.e., subtracting the coordinate-wise mean. Permutation invariance (Appendix A.5) is enforced by requiring the univariate function to be independent of the node index. In Theorem 4.2 (proof in Theorem A.10 of the Appendix), we present a permutation-invariant representation, related to DeepSets (Zaheer et al., 2017), that, in contrast to prior results, uses univariate functions.

---
**Permutation invariance v2**

**Lemma 4.2.** *Let* $m \in \mathbb{N}$ *and set* $M = 2m + 1$ *and let* $K \subseteq \mathbb{R}^d$ *be a compact set, then for any permutation-invariant continuous* $f : K^m \to \mathbb{R}$, *there exists continuous univariate functions* $\psi_q, \phi_{qp}, \varphi_r$ *such that*

$$f(x_1, \ldots, x_m) = F\left( \sum_{p=1}^{m} \varphi_1(x_p), \ldots, \sum_{p=1}^{m} \varphi_{2m+1}(x_p) \right), \quad x = (x_1, \cdots, x_m) \in \mathbb{R}^m.$$

*with* $F(y_1, \ldots, y_M) = \sum_{q=1}^{2M+1} \psi_q(\sum_{p=1}^{M} \phi_{qp}(y_p)), \quad y = (y_1, \cdots, y_M) \in \mathbb{R}^M.$
---

**General linear group** In Theorem 4.3 (see Section A and Theorem A.5), we have a weaker, but more general version of the invariant representation for the General Linear (GL) group, which represents a large class of groups, with $(i|j)$ the contractions of $GL(n)$.

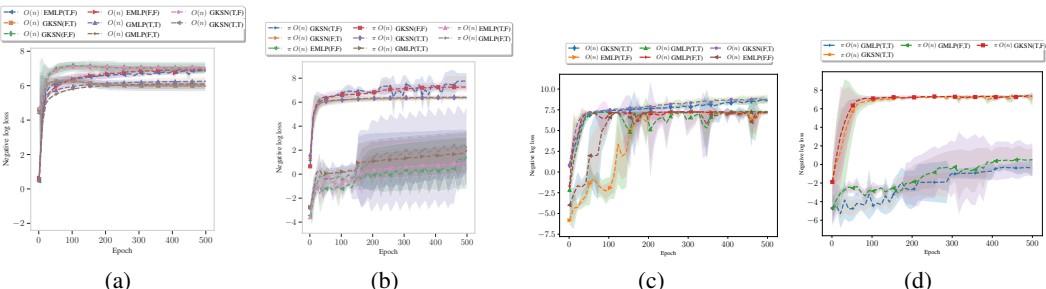

(a)      (b)      (c)      (d)

Figure 2: Plots visualize the negative log Huber loss $\uparrow$ over epochs. In parenthesis, the two flags indicate the use of node index $(T, *)$ or not $(F, *)$; the second flag the use of the linear $(*, T)$ (according to Equation 5) or quadratic $(*, F)$ version (according to Equation 4). a) Test performance of $O(n)$ invariant models for the LJ experiment with $n = 5$ and $m = 15$, while b) is the test performance of $O(n)$ and permutation invariant models. c) Test performance of various models for the Buckyball-Catcher dataset of MD22 where $\pi\, O(n)$ are the models that are invariant to rotation, reflection, and permutation, while d) for $O(n)$ invariant models to rotation and reflection on $\mathbb{R}^n$.

---

**$GL(n)$ invariance for polynomials**

**Theorem 4.3.** *For a $GL(n)$-invariant polynomial function $f(\boldsymbol{x}_1, \ldots, \boldsymbol{x}_m)\ :\ X^m \to \mathbb{R}$, with $X \subset \mathbb{R}^n$ a compact space, $f$ can be represented as $f(\boldsymbol{x}_1, \ldots, \boldsymbol{x}_m) = \sum_{q=1}^{2m^2+1} \psi_q(\sum_{i,j=1}^{m,m} \phi_{qij}((\boldsymbol{x}_i|\boldsymbol{x}_j)))$,*

---

## 5   GEOMETRIC KOLMOGOROV SUPERPOSITION NETWORKS (GKSN)

Finding the representation functions $\psi_q, \phi_{pq}$ is still a difficult non-linear optimization problem. To reduce the complexity of training, we consider a representation as a layer and allow the composition of multiple layers (Figure 1). The fundamental result from Equation 5 is that we can use univariate functions on invariant features. We consider a single layer of the Geometric Kolmogorov superposition networks (GKSN) as the composition of univariate functions $\phi_{pq}^{\ell}$ and the subsequent univariate functions $\psi_q^{\ell}$. With an abuse of notation and dropping the $\ell$ dependence on the functions, we write

$$\boldsymbol{z}_{\ell+1} = \overbrace{\boldsymbol{\Psi}}^{l \times k} \circ \overbrace{\boldsymbol{\Phi}^T}^{k \times m} (\boldsymbol{z}_\ell) + \overbrace{\boldsymbol{W}_\psi}^{l \times k'} \sigma(\overbrace{\boldsymbol{W}_\phi^T}^{k' \times m} \boldsymbol{z}_\ell), \qquad (6)$$

or if we compute the $i$-th element,

$$z_i^{\ell+1} = \underbrace{\sum_k \psi_{ik}\left(\underbrace{\sum_j \phi_{jk}(z_j^\ell)}_{\psi_{ik}(.)}\right)}_{\text{KST}} + \underbrace{\sum_k w_{ik}^\psi \sigma\left(\underbrace{\sum_j w_{ji}^\phi z_j^\ell}_{\phi_{jk}(.)}\right)}_{\text{Residue term}},$$

where $\circ$ is the function composition operator. The first term is the classical KST form, while the second is inspired by the newer forms (Table 1,(Laczkovich, 2021)) where univariate functions are combined linearly and where we use $\phi_q(x_p) = x_p, \psi(y) = \sigma(y)$. We, therefore, assume that the original function can be represented as the sum of two functions, the first with smooth but non-linear univariate functions, the second with the composition of a scaled non-linear function and the sum of linear functions. We further assume $\sigma$ to be a fixed, almost everywhere smooth, continuous, and almost linear to improve the training with multiple layers. The second path plays a role similar to the residual connection, which helps the training of the nonlinear univariate functions. Further, to reduce the number of parameters, we ignore $\psi_{ik}$ and set $l = k$, while $k'$ is set to a small value.

Table 2: Huber NLL ($\uparrow$, higher is better) for the LJ dataset on different dimensions ($n \in [3, 5]$) and number of nodes $m \in [4, 10, 15]$. Standard deviation as superscript, mean computed over 3 runs.

| LJ $m/n$ | $O(n)$ GKSN | EMLP | $\pi O(n)$ GKSN | $\pi O(n)$ GMLP |
|----------|-------------|------|-----------------|-----------------|
| 4/3 | $\mathbf{8.41}^{\pm 0.19}$ | $8.00^{\pm 0.12}$ | $7.88^{\pm 0.15}$ | $7.59^{\pm 0.14}$ |
| 10/3 | $\mathbf{7.10}^{\pm 0.16}$ | $6.76^{\pm 0.09}$ | $\mathbf{7.08}^{\pm 0.28}$ | $5.33^{\pm 0.18}$ |
| 10/5 | $7.15^{\pm 0.37}$ | $\mathbf{6.71}^{\pm 0.28}$ | $7.23^{\pm 0.41}$ | $3.72^{\pm 0.60}$ |
| 15/3 | $\mathbf{7.25}^{\pm 1.25}$ | $\mathbf{7.09}^{\pm 1.10}$ | $7.28^{\pm 1.17}$ | $3.92^{\pm 0.41}$ |
| 15/5 | $6.73^{\pm 0.18}$ | $6.56^{\pm 0.13}$ | $\mathbf{6.96}^{\pm 0.24}$ | $1.76^{\pm 1.33}$ |

## 6 EXPERIMENTAL EVALUATION

After presenting the experimental setup, we show the performance on representative datasets. To evaluate the representation power of the GKSN to model an invariant function to $E(3)$-symmetry action, we consider the task of training atomistic energy from atomic system configurations. We therefore considered the Lennard-Jones particle system (Section 6.2), Linear polymers (Appendix C.1), the MD17 (Section 6.2), and MD22 (Section 6.2) datasets. We also experiment with the use of GKSN for Lorentz symmetries, in particular, we study the Top jets stream classification (Section 6.2), Quark-Gluon tagging (Section 6.2), and symmetry discovery (Appendix C.2).

### 6.1 EXPERIMENTAL SETUP AND BASELINES

We compare different models to learn invariant functions from data, from both synthetic and real datasets. In the test, we normalize the output to the interval $[0, 1]$.

**Symmetries and Networks**  We denote by $O(n)$ the class of models with rotational and reflectional symmetry, and by $\pi$ the class of models with permutation symmetry. We mainly compare against the use of two-layer **MLP**-based models: the EMLP from (Villar et al., 2021)) and GMLP, based on our permutation invariant representation (Section A.2) or the linear representation (Appendix A.4). We implemented the **GKSN** model of Equation 6, where we use ReLU (Glorot et al., 2011) both as the basis for the GKSN non-linear functions ($\psi_q, \phi_{pq}$) and for the residual connection ($\sigma$). The name of the model contains two symbols $T$=True and $F$=False; the first boolean tells us if the node index is used as an additional $O(n)$ invariant feature. The effect of adding the index of the node is to emulate the non-permutation invariant function. The second boolean is used to show if the linear ($T$) (Equation 5) or quadratic ($F$) (Equation 4) feature is used. Therefore, $\pi O(n)$ GKSN($T, T$) is a permutation invariant model, where node index is used as a feature, where the number of features is linear in the number of nodes $m$.

**Invariant Features**  From Equation 5, the inner product is sufficient to approximate any invariant function. Therefore, by assuming that invariant features can be implicitly learnt during a training process with the inner product, we explore other invariant features as the input to improve expressivity. While the choice of invariant features is left heuristic, Equation 5 allows users to take a "shortcut" to improve the training efficiency and could potentially help inject better inductive bias to a model than the mere theory-based input whose effectiveness is also unknown. Concretely, we extend the invariant feature to include:

$$\|\boldsymbol{x}_i\|, \ \|\boldsymbol{y}_j\|, \ \|\boldsymbol{x}_i - \boldsymbol{y}_j\|, \ \langle \boldsymbol{x}_i, \boldsymbol{y}_j \rangle, \ \sqrt{\|\boldsymbol{x}_i\|^2 \|\boldsymbol{y}_j\|^2 - \langle \boldsymbol{x}_i, \boldsymbol{y}_j \rangle^2}$$

As additional invariant features, we optionally include the node index (first flag), and when present (experiments with MD17 and MD22), we also include the atom type. We have not explored alternative ways to embed the node's additional information as input to the network. The last term is also equivalent to $\|x \otimes y\|$ in $n = 3$ dimensions, with $\otimes$ the cross product. In Section F, we propose an ablation study on the effect of features on the representation power.

**Quadratic versus Linear features**  A consequence of Equation 5, with the associated theorem, is that the number of invariant features that we need is linear with the number of nodes. We nevertheless compare also with the quadratic version as in Equation 4.

Table 3: Huber NLL ↑ for the MD17 dataset (mean and standard deviation in parenthesis)

| Dataset (MD17) | $O(n)$ GKSN | EMLP | $\pi\,O(n)$ GKSN | $\pi\,O(n)$ GMLP |
|---|---|---|---|---|
| Aspirin | $\mathbf{6.44}^{\pm 0.10}$ | $5.62^{\pm 0.01}$ | $5.69^{\pm 0.02}$ | $4.73^{\pm 0.27}$ |
| Benzene | $\mathbf{7.66}^{\pm 0.08}$ | $5.93^{\pm 0.01}$ | $6.51^{\pm 0.17}$ | $5.64^{\pm 0.13}$ |
| Ethanol | $\mathbf{7.57}^{\pm 0.04}$ | $5.44^{\pm 0.01}$ | $6.09^{\pm 0.13}$ | $5.49^{\pm 0.03}$ |
| Malonaldehyde | $\mathbf{7.50}^{\pm 0.05}$ | $5.39^{\pm 0.01}$ | $5.85^{\pm 0.04}$ | $5.38^{\pm 0.04}$ |
| Naphthalene | $\mathbf{6.85}^{\pm 0.07}$ | $5.35^{\pm 0.00}$ | $5.72^{\pm 0.09}$ | $4.65^{\pm 0.76}$ |
| Salicylic | $\mathbf{6.96}^{\pm 0.09}$ | $5.62^{\pm 0.00}$ | $5.83^{\pm 0.10}$ | $5.17^{\pm 0.24}$ |
| Toluene | $\mathbf{7.05}^{\pm 0.13}$ | $5.68^{\pm 0.02}$ | $6.03^{\pm 0.10}$ | $5.40^{\pm 0.11}$ |
| Uracil | $\mathbf{7.54}^{\pm 0.08}$ | $5.65^{\pm 0.01}$ | $6.10^{\pm 0.11}$ | $5.52^{\pm 0.05}$ |

Table 4: Performance aggregated at the level of the model type for the MD22 dataset; the performance is the negative log of the Huber loss ↑ (mean and standard deviation in parentheses);

| Dataset (MD22) | $O(n)$ GKSN | EMLP | $\pi\,O(n)$ GKSN | $\pi\,O(n)$ GMLP |
|---|---|---|---|---|
| AT-AT-CG-CG | $\mathbf{8.02}^{\pm 0.14}$ | $7.61^{\pm 0.05}$ | $7.73^{\pm 0.05}$ | $0.82^{\pm 0.32}$ |
| AT-AT | $\mathbf{7.32}^{\pm 0.21}$ | $6.56^{\pm 0.01}$ | $6.62^{\pm 0.03}$ | $0.82^{\pm 0.40}$ |
| Ac-Ala3-NHMe | $\mathbf{5.77}^{\pm 0.07}$ | $5.57^{\pm 0.00}$ | $5.57^{\pm 0.01}$ | $1.48^{\pm 1.08}$ |
| DHA | $\mathbf{5.64}^{\pm 0.07}$ | $5.52^{\pm 0.00}$ | $5.50^{\pm 0.01}$ | $0.04^{\pm 0.82}$ |
| Buckyball-catcher | $\mathbf{8.85}^{\pm 0.24}$ | $7.27^{\pm 0.01}$ | $7.41^{\pm 0.07}$ | $0.21^{\pm 0.71}$ |
| Stachyose | $\mathbf{6.30}^{\pm 0.12}$ | $5.70^{\pm 0.01}$ | $5.73^{\pm 0.03}$ | $1.36^{\pm 1.42}$ |

## 6.2 Experiments

**Lennard-Jones** Lennard-Jones potential approximates inter-molecular pair interaction and models repulsive and attractive interactions. It captures key physical principles and it is widely used to model solid, fluid, and gas states. More details are in subsection D.1. Figure 2 show the test regression loss during training for a system with 15 nodes in 3 dimensional space. The loss is plotted on a negative log scale. We use the Huber loss that is quadratic if the error is less than 1, and linear if larger. The test loss for the $O(n)$ invariant model (Figure 2.(a)) is regular during training and all models seem to have similar results, while in Figure 2.(b) the performance of permutation invariant models have quite different behavior. The MLP-based models are more unstable, while GKSN-based models have a more regular performance. Table 2 summarizes the regression accuracy at test time for all the models. The permutation invariance reduces the performance, but more remarkably, on smaller systems.

**MD17** MD17 dataset contains samples from a long molecular dynamics trajectory of a few small organic molecules (Chmiela et al., 2017). For each molecule, we split into $8,000$ training and $200$ test configurations. In Table 3 we show the negative log of the Huber loss (negative log loss - NLL), aggregated over various model options, while in Table 11 we provide the test loss for each model. The test loss during training for $O(n)$ invariant models is stable, but reducing the number of features leads to lower performance, while GKSN shows better accuracy. On the other hand, the training for the permutation invariant models is less stable, and the overall performance reduces while keeping the model size smaller. Table 3 summarizes the performance of all models in the various atomic systems of MD17, the GKSN shows consistently better performance, even with a smaller network size.

**MD22** MD22 dataset (Chmiela et al., 2023) contains samples from molecular dynamics trajectories of four major classes of biomolecules, as proteins, lipids, carbohydrates, nucleic acids, and supramolecules. In MD22, the number of atoms ranges from 42 to 370. For each molecule, we split into $8,000$ training and $200$ test configurations. In Table 4 we show the NLL aggregated over various model options, while in Table 12 we provide detailed information on the performance. Figure 2.(c-d) show the Huber NLL at test time for the Ac-Ala3-NHMe molecule, with and without permutation invariance. The test loss in negative log scale at training for the $O(n)$ invariant models reported in Figure 2.(c) is stable for GKSN, while MLP-based models show more unstable training and lower performance. The training for the permutation invariant models in Figure 2.(d) is even less stable for the MLPs, leading to low accuracy. Table 4 summarizes the performance of all models in the various systems of MD22, the GKSN shows better performance, even when the size of the network is smaller.

Table 5: Top-tagging experimental results, including LGN (Bogatskiy et al., 2020), LorentzNet (Gong et al., 2022), and other baselines (Komiske et al., 2019), (Qu & Gouskos, 2020), Results for EMLP-$SO(1,3)^+$ and EKAN -$SO(1,3)^+$ are from (Hu et al., 2025); $^*$ Train on $10^4$ samples

| Model | Accuracy | AUC | $1/\epsilon_B(0.5)$ | $1/\epsilon_B(0.3)$ |
|---|---|---|---|---|
| ResNeXt | 0.936 | 0.9837 | $302^{\pm 5}$ | $1147 \pm 58$ |
| P-CNN | 0.930 | 0.9803 | $201^{\pm 4}$ | $759^{\pm 24}$ |
| PFN | 0.932 | 0.9819 | $247^{\pm 3}$ | $888^{\pm 17}$ |
| ParticleNet | **0.940** | **0.9858** | $\mathbf{397^{\pm 7}}$ | $\mathbf{1615^{\pm 93}}$ |
| EGNN | 0.922 | 0.9760 | $148^{\pm 8}$ | $540^{\pm 49}$ |
| LGN | 0.929 | 0.9640 | $124^{\pm 20}$ | $435^{\pm 95}$ |
| EMLP | 0.771* | - | - | - |
| EKAN | 0.769* | - | - | - |
| LorentzNet | **0.942** | **0.9868** | $\mathbf{498^{\pm 18}}$ | $\mathbf{2195^{\pm 173}}$ |
| GKSN | **0.940** | **0.9858** | $445^{\pm 28}$ | $1634^{\pm 328}$ |

Table 6: Quark-gluon tagging experimental results. The LorentzNet, EGNN, and LGN results are averaged over 6 runs, GKSN over 3.

| Model | Accuracy | AUC | $1/\epsilon_B(0.5)$ | $1/\epsilon_B(0.3)$ |
|---|---|---|---|---|
| ResNeXt | 0.821 | 0.8960 | 30.9 | 80.8 |
| P-CNN | 0.827 | 0.9002 | 34.7 | 91.0 |
| PFN | - | 0.9005 | $34.7^{\pm 0.4}$ | - |
| ParticleNet | **0.840** | **0.9116** | $\mathbf{39.8^{\pm 0.2}}$ | $\mathbf{98.6^{\pm 1.3}}$ |
| EGNN | 0.803 | 0.8806 | $26.3^{\pm 0.3}$ | $76.6^{\pm 0.5}$ |
| LGN | 0.803 | 0.8324 | 16.0 | 44.3 |
| LorentzNet | **0.844** | **0.9156** | $\mathbf{42.4^{\pm 0.4}}$ | $\mathbf{110.2^{\pm 1.3}}$ |
| GKSN | **0.839** | **0.9127** | $\mathbf{39^{\pm 0.4}}$ | $\mathbf{101^{\pm 3.5}}$ |

**Top Tagging**    Lorentz group $SO(1,3)^+$ is an important set of transformations in many physics problems. Top-tagging dataset is an open benchmark dataset (Kasieczka et al., 2019) with the task of classifying between top quark jets and background jets. It consists of 2M observations, each consisting of four-dimensional momentum of up to 200 particle jets. The classification task is Lorentz invariant, where the rotated or boosted input momentum belongs to the same category. The results in Table 5 show that the performance of GKSN is comparable with ParticleNet and LorentzNet.

**Quark-gluon tagging**    In the Quark-gluon tagging dataset (Komiske et al., 2019), the task consists of discriminating light-quark from gluon-initiated jets. The dataset consists of 2 million jets in total, where half are gluon jets and half are background jets. The Quark-gluon tagging classification task is modelled with a Lorentz invariant function. (Bogatskiy et al., 2020) The results in Table 6 show that the performance of GKSN is, also in this dataset, comparable with ParticleNet and LorentzNet.

**Limitations and reproducibility statement**    This paper aims to advance in the field of machine learning and scientific discovery. The proposed architecture provides a solid foundation with headroom for improvement; in future work, we will explore extensions of GKSN to further enhance performance. We use networks of a compatible size, selecting the model for each architecture between small, medium, and large, and use the selected architecture across the tasks (Appendix E.1).

## 7 CONCLUSIONS

We propose GKSN for invariant and equivariant function representation, which is based on a new representation for invariant functions to group actions. The theoretical results in Section 4, provide a considerable improvement over previous results (Villar et al., 2021), reducing the complexity from quadratic to linear. We further tested the performance and compared it with MLP-based architectures on an ideal physical system, the Lennard-Jones experiment, and on two real molecular datasets, the MD17 and the MD22 datasets, and two particle physics datasets. The performance of GKSN improves with respect to MLP, and further investigation will show if this architecture can be extended to implement machine learning interatomic potentials.

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

SUPPLEMENTARY MATERIAL OF GEOMETRIC KOLMOGOROV SUPERPOSITION REPRESENTATION OF GROUP INVARIANT FUNCTION FOR COMPUTATIONAL SCIENCE

## A    MAIN THEOREMS FOR THE KOLMOGOROV SUPERPOSITION THEOREM FOR INVARIANT AND EQUIVARIANT FUNCTIONS

We first recall the Kolmogorov - Arnold and Ostrand theorems.

**Theorem A.1.** *(Kolmogorov, 2009)  For any integer $m \geq 2$ there are continuous real functions $\phi_{qp}(x)$ on the close unit interval $E = [0, 1]$ such that each continuous real function $f(x_1, \cdots, x_m)$ on the $m$-dimensional unit cube $E^m$ is representable as*

$$f(x_1, \ldots, x_m) = \sum_{q=1}^{2m+1} \psi_q(\sum_{p=1}^{m} \phi_{qp}(x_p)),$$

*where $\psi_q$ are continuous functions.*

**Theorem A.2.** *(Ostrand, 1965) For $p = 1, 2, \cdots, m$, let $X^p$ be a compact metric space of finite dimension $d_p$, and let $n = \sum_{p=1}^{m} d_p$. There exist continuous functions $\phi_{qp} : X^p \to [0, 1]$, for $p = 1, \cdots, m$ and $q = 1, 2, \cdots, 2n + 1$, such that every continuous real function $f$ defined on $\prod_{p=1}^{m} X^p$ is representable in the form*

$$f(x_1, \ldots, x_m) = \sum_{q=1}^{2n+1} \psi_q(\sum_{p=1}^{m} \phi_{qp}(x_p)),$$

*where the functions $\psi_q$ are real and continuous.*

We also summarize the invariance representation from "Lemma 1", and "Proposition 8" of (Villar et al., 2021).

**Theorem A.3.** *(Villar et al., 2021) (First Fundamental Theorem of $O(n)$ and $O(1, n-1)$) Suppose a function $f(x_1, \ldots, x_m) : (\mathbb{R}^n)^m \to \mathbb{R}$ is an $O(n)$ or $O(1, n-1)$ continuous invariant scalar function. Then, $f$ can be represented as a continuous function of only scalar product of the input $x_i$. That is, there is a continuous function $g$ such that $f(x_1, \ldots, x_m) = g(X^T X) = g((\langle x_i, x_j \rangle)_{i,j=1}^m)$, with $\langle x_i, x_j \rangle = x^T \Lambda x$ the invariant inner scalar product with metrics $\Lambda = 1$ for $O(n)$ and $\Lambda = diag(1, -1, \ldots, -1)$ for $O(1, n-1)$.*

The propositions in (Villar et al., 2021) are based on the First Fundamental Theorem of $GL(V, n)$, the generalized linear group over a finite-dimensional vector space $V$ of dimension $n$ (Kraft & Procesi, 1996).

**Theorem A.4.** *(Kraft & Procesi, 1996) (First Fundamental Theorem for $\mathrm{GL}(V, n)$) The ring of invariants for the action of $\mathrm{GL}(V, n)$ on $V^p \oplus V^{*q}$ is generated by the invariants $(i \mid j)$:*

$$K[V^p \oplus V^{*q}]^{\mathrm{GL}(V,n)} = K[(i \mid j) \mid i = 1, \ldots, p, \ j = 1, \ldots, q].$$

**Invariants of vectors and covectors**    Here, we briefly recall the idea behind the theorem. The theorem is based on the concept of invariants of vectors and covectors. Let $V$ be a finite-dimensional $K$-vector space, for example $K = \mathbb{C}$ and $V = \mathbb{C}^n$. Consider the representation of $\mathrm{GL}(V)$ on the vector space

$$W := \underbrace{V \oplus \cdots \oplus V}_{p \text{ times}} \oplus \underbrace{V^* \oplus \cdots \oplus V^*}_{q \text{ times}} =: V^p \oplus V^{*q},$$

consisting of $p$ copies of $V$ and $q$ copies of its dual space $V^*$, given by

$$g(v_1, \ldots, v_p, \varphi_1, \ldots, \varphi_q) := (gv_1, \ldots, gv_p, g\varphi_1, \ldots, g\varphi_q)$$

where $g\varphi_i$ is defined by $(g\varphi_i)(v) := \varphi_i(g^{-1}v)$ and $g \in GL(V, n)$. This representation on $V^*$ is the *dual* representation of $\mathrm{GL}(V)$ on $V$, where the elements of $V$ are called *vectors*, while elements of the dual space $V^*$ are called *covectors*. We want to describe the invariants of $V^p \oplus V^{*q}$ under this action.

For every pair $(i, j)$, $i = 1, \ldots, p$, $j = 1, \ldots, q$, we define the bilinear function $(i \mid j)$ on $V^p \oplus V^{*q}$ by

$$(i \mid j) : (v_1, \ldots, v_p, \varphi_1, \ldots, \varphi_q) \mapsto (v_i \mid \varphi_j) := \varphi_j(v_i).$$

These functions are called *contractions*, and they are invariant to the actions $g \in GL(V, n)$:

$$(i \mid j)(g(v, \varphi)) = (g\varphi_j)(gv_i) = \varphi_j(g^{-1}gv_i) = (i \mid j)(v, \varphi).$$

The First Fundamental Theorem (sometimes referred as FFT) states that these functions generate the ring of invariants, i.e. polynomial functions on $V$. We first present a result about the universal representation theorem for $GL(n)$-invariant polynomial functions.

> **$GL(n)$ invariance for polynomials**
>
> **Theorem A.5.** *For a $GL(n)$-invariant polynomial function $f(\boldsymbol{x}_1, \ldots, \boldsymbol{x}_m) : X^m \to \mathbb{R}$, with $X \subset \mathbb{R}^n$ a compact space, $f$ can be represented as*
>
> $$f(\boldsymbol{x}_1, \ldots, \boldsymbol{x}_m) = \sum_{q=1}^{2m^2+1} \psi_q\left( \sum_{i,j=1}^{m,m} \phi_{qij}((\boldsymbol{x}_i|\boldsymbol{x}_j)) \right),$$

*Proof.* By Theorem A.4, $f$ is represented by a polynomial function $g$ whose input is $\{(i|j)\}_{i,j}$:

$$f(\boldsymbol{x}_1, \ldots, \boldsymbol{x}_m) = g(\{(i|j)\}_{i,j}).$$

Since $g$ is continuous, we can apply Theorem A.1 to represent $g$, which completes the proof. $\square$

In the following, we will focus on the action of some representative subgroups of $GL(n)$ in scientific discovery and prove that the representation theorem shown above could be generalized to continuous (not necessarily polynomial) functions invariant or equivariant to the actions of the subgroups. We note that while we are aware that the result of (Ismailov, 2008) can further generalize some of the following claims to the case of non-continuous functions, we will leave this generalization for future research, especially for the case of learning non-continuous invariant and equivariant functions.

## A.1 INVARIANTS FOR SPECIFIC METRIC

The invariants for specific metrics or symmetric groups are

- Euclidean; Poincare
$$(i|j) = \langle \boldsymbol{x}_i, \boldsymbol{x}_j \rangle = \boldsymbol{x}_i^T \boldsymbol{x}_j$$

- Minkowski
$$(i|j) = \langle \boldsymbol{x}_i, \boldsymbol{x}_j \rangle = \boldsymbol{x}_i^T \boldsymbol{\Lambda} \boldsymbol{x}_j$$
with $\boldsymbol{\Lambda}_{ij} = 1/2(1 - 21_{i-1})\delta_{ij}$, i.e. $\boldsymbol{\Lambda}_{11} = -1$ and $\boldsymbol{\Lambda}_{ii} = 1, i > 1$

- $GL(V = \mathbb{R}^n)$
$$(i|j) = \langle \boldsymbol{x}_i, \boldsymbol{x}_j \rangle = \boldsymbol{x}_i^T \boldsymbol{A} \boldsymbol{x}_j$$

## A.2 $S_n$- PERMUTATION INVARIANCE

**Lemma A.6.** *(Permutation invariance) Suppose we have continuous real functions $\psi_q, \phi_q : \mathbb{R} \to \mathbb{R}$ for $\forall q \in [2m + 1]$. Then, the following function is invariant to the action of the permutation group:*

$$f(x_1, \ldots, x_m) = \sum_{q=1}^{2m+1} \psi_q\left( \sum_{p=1}^{m} \phi_q(x_p) \right), \quad x = (x_1, \cdots, x_m) \in \mathbb{R}^m.$$

*Proof.* Since the decomposition requires the output of the function to not change after a generic permutation $\pi$ of the input, then

$$\sum_{q=1}^{2m+1} \psi_q\left( \sum_{p=1}^{m} \phi_{qp}(x_p) \right) = \sum_{q=1}^{2m+1} \psi_q\left( \sum_{p=1}^{m} \phi_{qp}(x_{\pi(p)}) \right)$$

to be true, it is sufficient to drop the dependence of $\phi_{qp}$ on the node index $p$. $\square$

| | |
|---|---|
| Generalized Linear | $\mathrm{GL}(V, n) = \{M \in \mathbb{R}^{n \times n} : M^\top M = A, \det(A) \neq 0\},$ |
| | $M(v_1, \cdots, v_n) = (M\, v_1, \cdots, M\, v_n)$ |
| Orthogonal | $\mathrm{O}(n) = \{Q \in \mathbb{R}^{n \times n} : Q^\top Q = Q\, Q^\top = I_n\},$ |
| | $Q(v_1, \cdots, v_n) = (Q\, v_1, \cdots, Q\, v_n)$ |
| Rotation | $\mathrm{SO}(n) = \{Q \in \mathbb{R}^{n \times n} : Q^\top Q = Q\, Q^\top = I_n,\ \det(Q) = 1\}$ |
| | $Q(v_1, \cdots, v_n) = (Q\, v_1, \cdots, Q\, v_n)$ |
| Translation | $\mathrm{T}(n) = \{w \in \mathbb{R}^n\}$ |
| | $w(v_1, \cdots, v_n) = (v_1 + w, \cdots, v_n + w)$ |
| Euclidean | $\mathrm{E}(n) = \mathrm{T}(n) \otimes \mathrm{O}(n)$ |
| | $(w, Q)(v_1, \cdots, v_n) = (Q\, v_1 + w, \cdots, Q\, v_n + w)$ |
| Lorentz | $\mathrm{O}(1, n-1) = \{Q \in \mathbb{R}^{n \times n} : Q^\top \Lambda\, Q = \Lambda,\ \Lambda = \mathrm{diag}([-1, 1, \ldots, 1])\}$ |
| | $(w, Q)(v_1, \cdots, v_n) = (Q\, v_1 + w, \cdots, Q\, v_n + w)$ |
| Poincaré | $\mathrm{IO}(1, d) = \mathrm{T}(n) \otimes \mathrm{O}(1, n-1)$ |
| | $(w, Q)(v_1, \cdots, v_n) = (Q\, v_1 + w, \cdots, Q\, v_n + w)$ |
| Permutation | $\mathrm{S}_n = \{\sigma : [n] \to [n] \text{ bijective function}\}$ |
| | $\sigma(v_1, \ldots, v_n) = (v_{\sigma(1)}, \ldots, v_{\sigma(n)})$ |

Table 7: Similar to (Villar et al., 2021), we summarize here the more important symmetries we are considering. Groups ($G$) and the associated actions $g \in G$ of the groups on the elements of the vector space $\boldsymbol{v} = (v_1, \ldots, v_n) \in V$.

**Remark A.7.** *We note that while the expression looks similar to KAT, it is not known whether the above expression is universal for arbitrary permutation-invariant functions.*

### A.3 PERMUTATION INVARIANCE AND ITS CONNECTION TO DEEPSET

We present two theorems that connect DeepSet and KAT. (Zaheer et al., 2017) proposes a connection to KAT using high-dimensional functions; this theorem has been extended in (Amir et al., 2023) by considering linear functions.

**Theorem A.8** (**Theorem 7 (Zaheer et al., 2017)**). *(DeepSet Permutation Invariant representation) Let $f : [0, 1]^m \to \mathbb{R}$ be an arbitrary multivariate continuous function iff it has the representation*

$$f(x_1, ..., x_m) = \rho\left(\sum_{p=1}^m \phi(x_p)\right) \tag{7}$$

*with continuous outer and inner functions $\rho : \mathbb{R}^{2m+1} \to \mathbb{R}$ and $\phi : \mathbb{R} \to \mathbb{R}^{2m+1}$. The inner function $\phi$ is independent of the function $f$.*

**Corollary A.9** (**Corollary 6.1 (Amir et al., 2023)**). *Let $m, d \in \mathbb{N}$ and set $M = 2md + 1$. Let $\sigma : \mathbb{R} \to \mathbb{R}$ be an analytic non-polynomial function. Let $K \subseteq \mathbb{R}^d$ be a compact set. Then there exist $\boldsymbol{A} \in \mathbb{R}^{M \times d}, \boldsymbol{b} \in \mathbb{R}^M$ such that for any continuous permutation-invariant $f : K^m \to \mathbb{R}$, there exists a continuous $F : \mathbb{R}^M \to \mathbb{R}$ such that*

$$f(\boldsymbol{X}) = F\left(\sum_{p=1}^m \sigma(\boldsymbol{A}\boldsymbol{x}_p + \boldsymbol{b})\right), \quad \forall \boldsymbol{X} = (\boldsymbol{x}_1, \ldots, \boldsymbol{x}_m) \in K^m. \tag{8}$$

We notice that the feature used in Theorem A.6 have similarity with the features generated in Theorem A.9, therefore we propose the following lemma.

---

**Permutation invariance v2**

**Lemma A.10.** *Let $m \in \mathbb{N}$ and set $M = 2m + 1$ and let $K \subseteq \mathbb{R}^d$ be a compact set, then for any continuous permutation-invariant $f : K^m \to \mathbb{R}$, there exists continuous univariate functions $\psi_q, \phi_{qp}, \varphi_r$ such that*

$$f(x_1, \ldots, x_m) = F(\sum_{p=1}^{m} \varphi_1(x_p), \ldots, \sum_{p=1}^{m} \varphi_{2m+1}(x_p)), \;\; x = (x_1, \cdots, x_m) \in \mathbb{R}^m.$$

*with*

$$F(y_1, \ldots, y_M) = \sum_{q=1}^{2M+1} \psi_q(\sum_{p=1}^{M} \phi_{qp}(y_p)), \;\; y = (y_1, \cdots, y_M) \in \mathbb{R}^M.$$

---

*Proof.* The proof is based on Theorem A.9 setting $d = 1$ and building the functions

$$\varphi_r(x_p) = \sigma(\boldsymbol{A}_r x_p + \boldsymbol{b}_r)$$

where now $\boldsymbol{A}, \boldsymbol{b}$ are vectors of dimension $M$. Then apply KAT Theorem A.1 to the function $F(y_1, \ldots, y_M)$, remembering that the image of a compact set from a continuous and bounded function is compact. $\qquad\square$

Compared to Theorem A.6, this version, while stronger, requires $\approx M^3$ applications of univariate functions.

### A.4 $O(n)$-INVARIANCE

We here consider the permutation group that acts on the input $(\boldsymbol{x}_1, \ldots, \boldsymbol{x}_m)$ and present the architecture invariant to the action of the orthogonal group.

---

**$O(n)$ invariance - v1**

**Theorem A.11.** *For a continuous function invariant to the action of $O(n)$ $f(\boldsymbol{x}_1, \ldots, \boldsymbol{x}_m) :$ $X^m \to \mathbb{R}$, with $X \subset \mathbb{R}^n$ a compact space, it can be represented as*

$$f(\boldsymbol{x}_1, \ldots, \boldsymbol{x}_m) = \sum_{q=1}^{2m^2+1} \psi_q(\sum_{i,j=1}^{m,m} \phi_{qij}(\langle \boldsymbol{x}_i, \boldsymbol{x}_j \rangle)),$$

---

*Proof.* We first apply Theorem A.3 to get an invariant representation $f = g((\langle x_i, x_j \rangle)_{i,j=1}^{m})$ and apply Theorem A.1 for $g$ to get a KAT representation, which completes the proof. $\qquad\square$

The above invariant representation takes a high computational cost. In the following we give one computationally efficient model. The detail is also described in Appendix B.

---

**$O(n)$ invariance - v2**

**Theorem A.12.** *For a continuous function invariant to the action of $O(n)$ $f(\boldsymbol{x}_1, \ldots, \boldsymbol{x}_m) :$ $X^m \to \mathbb{R}$, with $X \subset \mathbb{R}^n$ a compact space, it can be represented as*

$$f(\boldsymbol{x}_1, \ldots, \boldsymbol{x}_m) = \sum_{q=1}^{2mn+1} \psi_q \left( \sum_{i=1,j=1}^{m,n} \phi_{qij}(\langle \boldsymbol{x}_i, \boldsymbol{y}_j \rangle) + \sum_{i=1,j=1}^{n,n} \phi'_{qij}(\langle \boldsymbol{y}_i, \boldsymbol{y}_j \rangle) \right),$$

*where $\boldsymbol{y}_j^q = \alpha_j^q(\boldsymbol{x}_1, \ldots, \boldsymbol{x}_m) = \sum_{p=1}^{m} \alpha_p^j \boldsymbol{x}_p$, with $\boldsymbol{y}_j^q$ a linear combination of $\{\boldsymbol{x}_p\}$ with scalars $\alpha_p$ such that $span(\{\boldsymbol{y}_j^q\}_{j=1}^{n}) = \mathbb{R}^n$.*

---

*Proof.* The proof is based on the use of Theorem A.3 , Theorem A.19 and Theorem A.1. Since we define $y_j$ as linear combination of $x_p$ then also $\langle x_p, y_j \rangle$ and $\langle y_p, y_j \rangle$ are invariant to rotation, e.g. $\langle Rx_p, y_j' \rangle = \langle Rx_p, \sum \alpha_i Rx_i \rangle = \langle Rx_p, R\sum \alpha_i x_i \rangle = \langle Rx_p, Ry_j \rangle = \langle x_p, y_j \rangle.$ □

As a corollary, we get the following further compact form when input vectors span $\mathbb{R}^n$. The derivation is done in a manner similar to Theorem A.12 except applying Theorem A.20 instead of Theorem A.19:

---

**$O(n)$ invariance - v3**

**Corollary A.13.** *(same as Theorem 4.1) Suppose that* $\mathrm{span}(\{\boldsymbol{x}_j\}_{j=1}^n) = \mathbb{R}^n$. *Then, a continuous function invariant to the action of* $O(n)$ $f(\boldsymbol{x}_1, \ldots, \boldsymbol{x}_m) : X^m \to \mathbb{R}$, *with* $X \subset \mathbb{R}^n$ *a compact space, can be represented as*

$$f(\boldsymbol{x}_1, \ldots, \boldsymbol{x}_m) = \sum_{q=1}^{2mn+1} \psi_q \left( \sum_{i=1,j=1}^{m,n} \phi_{qij}(\langle \boldsymbol{x}_i, \boldsymbol{x}_j \rangle) \right).$$

---

### A.5  $O(n)$ AND PERMUTATION INVARIANCE

We further consider the permutation group action to the input $(\boldsymbol{x}_1, \ldots, \boldsymbol{x}_m)$ and present the architecture invariant to the action of the permutation group.

**Corollary A.14.** *($O(n)$ and $S_n$ permutation invariance - v1) The following function is invariant to the action of the permutation group and the orthogonal group $O(n)$:*

$$f(\boldsymbol{x}_1, \ldots, \boldsymbol{x}_m) = \sum_{q=1}^{2mn+1} \psi_q \left( \sum_{i=1,j=1}^{m,m} \phi_q(\langle \boldsymbol{x}_i, \boldsymbol{x}_j \rangle) \right).$$

*Proof.* We based this result on Theorem A.11 and Theorem A.6, by removing the dependence on the node index. □

**Remark A.15.** *We note that while the expression looks quite similar to KAT in appearance, it is not known whether the above expression is universal for arbitrary $O(n)$ permutation invariant functions.*

### A.6  $O(n)$-EQUIVARIANCE

We have the corresponding equivariant version.

---

**$O(n)$ equivariance - v1**

**Theorem A.16.** *Suppose that $\mathrm{span}(\{\boldsymbol{x}_j\}_{j=1}^n) = \mathbb{R}^n$. For a continuous function equivariant to the action of $O(n)$ $f(x_1, \ldots, x_m) : X^m \to X$, with $X \subset \mathbb{R}^n$ compact space, it can be represented as*

$$f(\boldsymbol{x}_1, \ldots, \boldsymbol{x}_m) = \sum_{k=1}^{m} \sum_{q=1}^{2mn+1} \psi_q^k \left( \sum_{i=1,j=1}^{m,m} \phi_{qij}^k(\langle \boldsymbol{x}_i, \boldsymbol{x}_j \rangle) \right) \boldsymbol{x}_k.$$

---

*Proof.* We use (Proposition 4) from (Villar et al., 2021) to get an equivariant form multiplied with $O(n)$-invariant functions and apply Theorem A.11 to the respective invariant functions. □

Similar results can be obtained for the representation from Theorem A.12 or Theorem A.13.

It is also possible to show that we can use the gradients of invariant functions to build a generic equivariant function, in particular, if $f(\boldsymbol{x}, \ldots, \boldsymbol{x}_m)$ is invariant, then $\nabla_{\boldsymbol{x}_i} f(\boldsymbol{x}, \ldots, \boldsymbol{x}_m)$ is equivariant, as it is $\sum_{i=1}^{m} \alpha_i \nabla_{\boldsymbol{x}_i} f(\boldsymbol{x}, \ldots, \boldsymbol{x}_m)$. Extending the previous results with these forms is easy when $f$ is decomposed according to Theorem A.11, Theorem A.12 or Theorem A.13.

## A.7   $O(n)$- EQUIVARIANCE AND PERMUTATION INVARIANCE

We have the corresponding equivariant and permutation invariant versions.

**Corollary A.17.** *($O(n)$ equivariance and permutation invariance - v1) The following function is invariant to the action of the permutation group and the orthogonal group $O(n)$:*

$$f(\boldsymbol{x}_1, \ldots, \boldsymbol{x}_m) = \sum_{i=1}^{m} \sum_{q=1}^{2mn+1} \psi_q \left( \sum_{j=1}^{m} \phi_q(\langle \boldsymbol{x}_i, \boldsymbol{x}_j \rangle) \right) \boldsymbol{x}_i.$$

*Proof.* We based this result on Theorem A.16 and Theorem A.6. □

## A.8   MAPPING INVARIANT FEATURES

**Lemma A.18.** *Suppose that we have $\boldsymbol{X} \in \mathbb{R}^{m \times n}$ and $\boldsymbol{Y} \in \mathbb{R}^{k \times n}$ with $\rho(\boldsymbol{Y}) = n, n \leq k$ then*

$$\boldsymbol{X}\boldsymbol{Y}^T(\boldsymbol{Y}\boldsymbol{Y}^T)^\dagger \boldsymbol{Y}\boldsymbol{X}^T = \boldsymbol{X}\boldsymbol{X}^T,$$

*where $\rho(X)$ is the matrix rank and $^\dagger$ is the pseudo-inverse.*

*Proof.* The equality follows from these properties:

$$\boldsymbol{Y} = \boldsymbol{V}\boldsymbol{\Lambda}\boldsymbol{U}, \quad \boldsymbol{V}^T\boldsymbol{V} = \boldsymbol{I}_k, \quad \boldsymbol{U}^T\boldsymbol{U} = \boldsymbol{I}_n = \boldsymbol{U}\boldsymbol{U}^T,$$

$$(\boldsymbol{Y}\boldsymbol{Y}^T)^\dagger = (\boldsymbol{V}\boldsymbol{\Lambda}\boldsymbol{\Lambda}^T\boldsymbol{V}^T)^\dagger = \boldsymbol{V}(\boldsymbol{\Lambda}\boldsymbol{\Lambda}^T)^\dagger\boldsymbol{V}^T, \quad \boldsymbol{Y}^T = \boldsymbol{U}^T\boldsymbol{\Lambda}^T\boldsymbol{V}^T,$$

$$\boldsymbol{Y}^T(\boldsymbol{Y}\boldsymbol{Y}^T)^\dagger\boldsymbol{Y} = \boldsymbol{U}^T\boldsymbol{\Lambda}^T\boldsymbol{V}^T\boldsymbol{V}(\boldsymbol{\Lambda}\boldsymbol{\Lambda}^T)^\dagger\boldsymbol{V}^T\boldsymbol{V}\boldsymbol{\Lambda}\boldsymbol{U} = \boldsymbol{U}^T\boldsymbol{\Lambda}^T(\boldsymbol{\Lambda}\boldsymbol{\Lambda}^T)^\dagger\boldsymbol{\Lambda}\boldsymbol{U} = \boldsymbol{I}_n.$$

□

**Theorem A.19.** *(Correlation matrix representation) Given $\boldsymbol{x}_1, \ldots, \boldsymbol{x}_m \in \mathbb{R}^n$ and a set of points $\boldsymbol{y}_1, \ldots, \boldsymbol{y}_k \in \mathbb{R}^n$, such that $\rho(\boldsymbol{y}_1, \ldots, \boldsymbol{y}_k) = n$, there is an map from $\mathcal{A}$ to $\mathcal{B}$, where :*

- *$\mathcal{B} = \{\langle \boldsymbol{x}_i, \boldsymbol{x}_j \rangle\}_{i,j=1}^{m,m}$, with a total number of variable equal to $m^2$*

- *$\mathcal{A} = \{\langle \boldsymbol{x}_i, \boldsymbol{y}_j \rangle\}_{i,j=1}^{m,k} \bigcup \{\langle \boldsymbol{y}_i, \boldsymbol{y}_j \rangle\}_{i,j=1}^{k,k}$ with a total number of variable equal to $mk + k^2$*

*Proof.* Define $\boldsymbol{X} = (\boldsymbol{x}_1, \ldots, \boldsymbol{x}_m)^T \in \mathbb{R}^{m \times n}$ and $\boldsymbol{Y} = (\boldsymbol{y}_1, \ldots, \boldsymbol{y}_k)^T \in \mathbb{R}^{k \times n}$ then

$$\boldsymbol{X}\boldsymbol{X}^T = \{\langle \boldsymbol{x}_i, \boldsymbol{x}_j \rangle\}_{i,j=1}^{m,m}, \quad \boldsymbol{X}\boldsymbol{Y}^T = \{\langle \boldsymbol{x}_i, \boldsymbol{y}_j \rangle\}_{i,j=1}^{m,k}, \text{ and } \boldsymbol{Y}\boldsymbol{Y}^T = \{\langle \boldsymbol{y}_i, \boldsymbol{y}_j \rangle\}_{i,j=1}^{k,k}.$$

We then apply Theorem A.18 to yield $\boldsymbol{X}\boldsymbol{X}^T = \boldsymbol{X}\boldsymbol{Y}^T(\boldsymbol{Y}\boldsymbol{Y}^T)^\dagger\boldsymbol{Y}\boldsymbol{X}^T$. Notice that $\boldsymbol{Y}\boldsymbol{X}^T = (\boldsymbol{X}\boldsymbol{Y}^T)^T$, and therefore we have the result. Another way to see it is that $\boldsymbol{C} = \boldsymbol{X}\boldsymbol{Y}^T, \boldsymbol{D} = \boldsymbol{Y}\boldsymbol{Y}^T, \boldsymbol{A} = \boldsymbol{X}\boldsymbol{X}^T$, then according to Theorem A.18, we have $\boldsymbol{C}\boldsymbol{D}^\dagger\boldsymbol{C}^T = \boldsymbol{A}$. □

We define $\boldsymbol{Y}$ as a subset of $\boldsymbol{X} \in \mathbb{R}^{m \times n}$ of size $k$, then it is a matrix of dimension $k \times n$, which we ask to have rank $n$. We can then say,

**Corollary A.20.** *(Special case - Subset) If $\boldsymbol{Y} = \boldsymbol{X}[: n]$, with $n \leq k$, $\rho(\boldsymbol{Y}) = n$, $\boldsymbol{X} \in \mathbb{R}^{m \times n}$, $m \leq k \leq n$, then there is an invertible map between these two sets:*

- *$\mathcal{B} = \{\langle \boldsymbol{x}_i, \boldsymbol{x}_j \rangle\}_{i,j=1}^{m,m}$, with a total number of variable equal to $m^2$*

- *$\mathcal{A}' = \{\langle \boldsymbol{x}_i, \boldsymbol{x}_j \rangle\}_{i,j=1}^{m,k}$, if $\boldsymbol{y}_j = \boldsymbol{x}_j$, with a total number of variable equal to $mk$,*

*Proof.* We use Theorem A.19 and notice that $\boldsymbol{Y}\boldsymbol{Y}^T$ can be derived from $\boldsymbol{Y}\boldsymbol{Y}^T = \boldsymbol{X}[: n]\boldsymbol{X}[: n]^T$, which are included in the previous features. For the reverse map, we simply ignore $\{\langle \boldsymbol{x}_i, \boldsymbol{x}_j \rangle\}_{i=1,j=k+1}^{m,m} = \mathcal{B} \bigcap \mathcal{A}'$ □

## A.9 COMPUTATIONAL VALIDATION OF THE MAIN THEOREM

There is one step in our theorem that creates concern. This step is as follows: once we change the basis for our data, we build the basis from the data itself. We now prove with a simple Python code that this is the case.

```python
# some help functions
rot_gen = lambda n: np.linalg.svd(np.random.randn(n,n))[0]
basis = lambda X: X[:n,:]
corr = lambda X: X @ X.T
inv = lambda X,Y: X @ Y.T
rot = lambda X,R: X @ R
#set the seed; it can be removed or changes
np.random.seed(42)
# the problem's dimension, can be changed, but m>=n
m,n = 5,3
# this is my data
X = np.random.randn(m,n)
# the correlation matrix of the data, which is an invariant feature
C1 = corr(X)
# we build a basis that depends on the input
Y = basis(X)
# compujte invariant features
Z1 = inv(X,Y)
# compute the correlation of the new features
D1 = corr(Z1)
# some rotation
R = rot_gen(n)
# apply the rotation to the input
X = rot(X, R)
# rebuild the basis
Y = basis(X)
# compute the invariant features
Z2 = inv(X,Y)
# compute the correlation with the new invariant features
D2 = corr(Z2)
# Question: is the correlation matrix before and after the same (we know
    is the same):
print(np.linalg.norm(C1 - C2))
# Result: 1.934545700657722e-15 (yes, numerically the same)
# Question: is the correlation matrix with the invariant feature the same
    before and after (they should)
print(np.linalg.norm(D1 - D2))
# Result: 9.407543438562363e-15 (yes, numerically the same)
# Question: are the invariant feature the same, before and after the
    rotation (they better be)?
print(np.linalg.norm(Z1 - Z2))
# Result: 1.4220500840710913e-15 (yes, numerically the same)
```
Listing 1: Python code to validate the contribution.

## A.10 COMPUTATIONAL VALIDATION OF THEOREM A.18

```python
import numpy as np
from numpy.linalg import norm
np.random.seed(42)
# the problem's dimension, can be changed, but m>=n
m,n = 15,3
k = n+2
# create the two matricies
X = np.random.randn(m,n)
Y = np.random.randn(k,n)
# Verify Theorem A.14
print(norm(X @ Y.T @ np.linalg.pinv(Y @ Y.T) @ Y @ X.T - X@X.T))
```

Table 8: Huber NLL for the Linear Polymer dataset, with $a_i = 0$ on different dimensions $(3, 5)$ and different number of nodes $4, 10, 15$.

| LinPoly-1 | $O(n)$ GKSN | EMLP | $\pi\,O(n)$ GKSN | $\pi\,O(n)$ GMLP |
|---|---|---|---|---|
| m4/n3 | 10.85 | 11.74 | 9.07 | 9.29 |
|  | 0.52 | 0.18 | 0.40 | 0.33 |
| m10/n3 | 8.93 | 9.08 | 7.36 | 6.40 |
|  | 0.33 | 0.36 | 0.32 | 0.25 |
| m10/n5 | 9.22 | 9.06 | 7.41 | 5.97 |
|  | 0.10 | 0.16 | 0.14 | 0.12 |
| m15/n3 | 7.99 | 7.98 | 6.91 | 4.82 |
|  | 0.34 | 0.24 | 0.38 | 0.51 |
| m15/n5 | 7.99 | 7.81 | 6.76 | 4.18 |
|  | 0.33 | 0.16 | 0.39 | 0.78 |

```
12  # Result: 1.2816111681783468e-14
```

Listing 2: Python code to validate the contribution.

## B  COMPLEXITY

The representation complexity of Equation 4 is $O(m^4)$, which is quite larger than the complexity we have if we apply KAT directly to the coordinates of the nodes, i.e. $O(m^2n^2)$, which ignores the symmetries of the problem. However, in Equation 5, we show that we can represent the invariant function $f$ with complexity $O(m^2n^2)$, thus similar to the non-invariant KAT.

## C  ADDITIONAL EXPERIMENTS

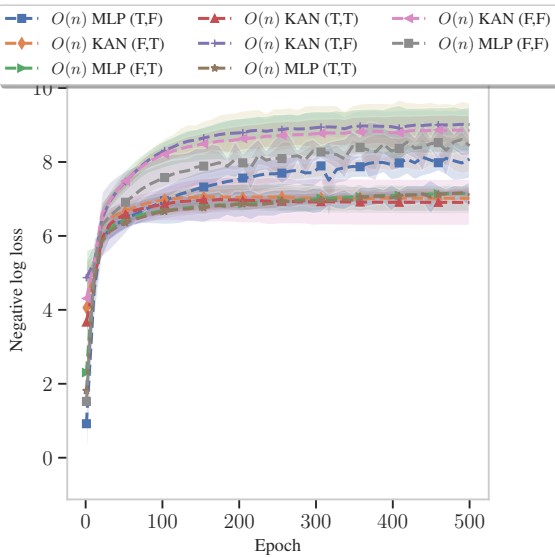

Figure 3: Test performance (Negative log Huber Loss) of various models for the linear polymers. $O(n)$ is the model that is invariant to rotation and reflection on $\mathbb{R}^n$, while $\pi$ is the permutation invariant model. In parenthesis, the two flags indicate if the model includes the node index and the second if the features are linear or quadratic in the number of nodes.

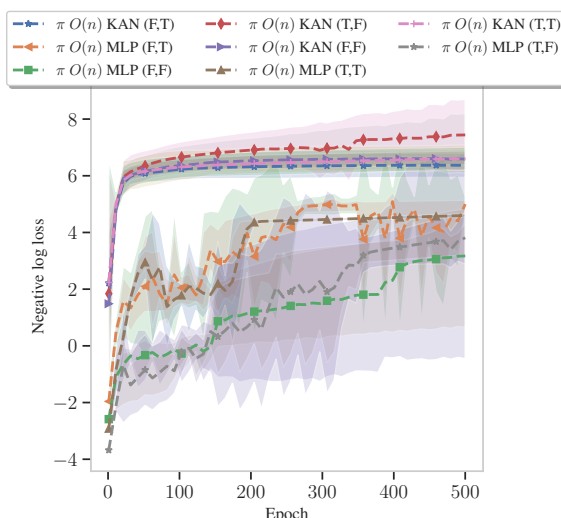

Figure 4: Test performance (Negative log Huber Loss) of various models for the linear polymers. $O(n)$ is the model that is invariant to rotation and reflection on $\mathbb{R}^n$, while $\pi$ is the permutation invariant model. In parentheses, the two flags indicate if the model includes the node index and the second if the features are linear or quadratic in the number of nodes.

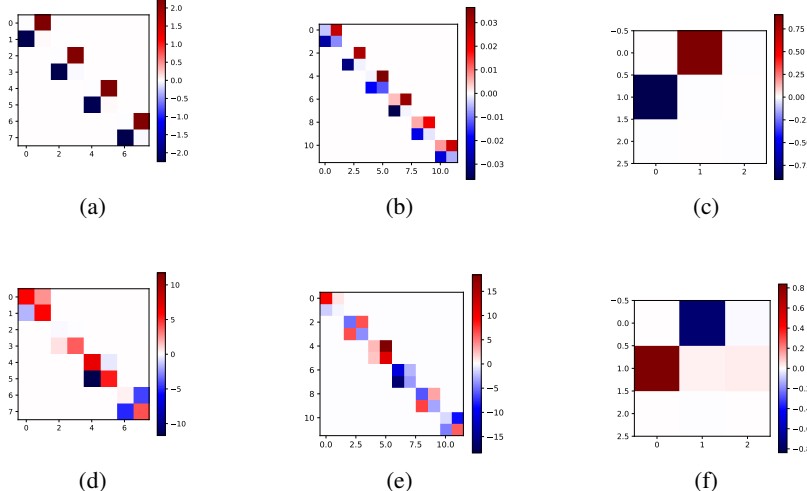

Figure 5: Visualization of discovered symmetries. From left to right: 2-body, 3-body and discrete permutation tasks; the top plots are for the LiGAN method and the bottom plots for LieGAN-GKSN.

### C.1 LINEAR POLYMER EXPERIMENTS

Linear polymers are chain molecules composed of repeating structural units (monomers) linked together sequentially. Linear polymers exhibit flexibility and thermoplastic behavior. Examples include polyethylene (PE), polyvinyl chloride (PVC), and polystyrene (PS), and find applications in packaging, textiles, and plastic films due to their ease of processing, recyclability, and ability to be melted and reshaped. Figure 3 and Figure 4 show the performance with $O(n)$ symmetry and with additionally permutation symmetry. Additional details in subsection D.2.

### C.2 SYMMETRY DISCOVERY

To test symmetry discovery, we extend the LiGAN (Yang et al., 2023) from (Yang et al., 2023) with GKSN and test on: 1) N-body trajectories; 2) Discrete Rotations. The learned metric matrices are shown in Figure 5, where HNN from (Greydanus et al., 2019) was used by (Yang et al., 2023).

Table 9: Huber NLL for the LJ-2 dataset

| LJ (2) | $O(n)$ GKSN | EMLP | $\pi\, O(n)$ GKSN | $\pi\, O(n)$ GMLP |
|--------|-------------|------|-------------------|-------------------|
| m4/n3  | 9.54 | 8.52 | 9.35 | 8.56 |
|        | 0.82 | 0.43 | 0.62 | 0.39 |
| m10/n3 | 8.66 | 8.22 | 8.49 | 5.73 |
|        | 0.66 | 0.61 | 0.74 | 0.25 |
| m10/n5 | 7.52 | 7.02 | 7.19 | 4.84 |
|        | 0.27 | 0.10 | 0.32 | 0.19 |
| m15/n3 | 9.45 | 9.35 | 9.89 | 3.91 |
|        | 1.32 | 1.43 | 2.11 | 0.79 |
| m15/n5 | 6.66 | 6.47 | 6.74 | 2.36 |
|        | 0.23 | 0.27 | 0.25 | 1.33 |

**N-Body Trajectory**    The task is to discover symmetry from 2-body and 3-body trajectory prediction

**Discrete Rotation Invariant Regression**    The task is to discover symmetry for a discrete rotation.

## D    EXPERIMENTS

The LJ task aims to train the energy from atomistic configurations (3d conformations), where the configurations are generated using random positions, followed by energy minimization.

The MD task aims at training the atomistic energy from DFT energy computations. The MD datasets are generated using molecular dynamics, therefore, they represent more realistic configurations of the atomic systems.

### D.1    LENNARD-JONNES

For the Lennard-Jonnes (LJ) experiments, we generate $m$ particles in $n$ dimensional space. The interaction between particles is described by the LJ potential,

$$U_{\mathrm{LJ}}(r) = f((a/r)^{12} - (a/r)^6)$$

where $r$ is the distance between two particles and $a$ is a parameter that defines the minimum energy of the interaction, while $f(x) = x + \sum_{l=1}^{3} a_l \sin(w_l x)$, with $a_1 = 1, a_2 = .3, a_3 = .1, w_1 = 11, w_2 = 30, w_3 = 50$ (or $a_1 = a_2 = a_3 = 0$), is an oscillatory term. After generating the particles, we perform an energy minimization step to relax the system towards a lower energy state, avoiding large energy contributions caused by the random initialization of the particle positions.

### D.2    LINEAR POLYMERS

As an additional experiment, we consider linear polymers of size $m$. The particles are connected to the previous and the following particle by a bond. The interaction between the bond depends quadratically on the difference between the current distance and the desired distance,

$$U_{\mathrm{bond}}(r) = f(\|d - \hat{d}\|^2) + U_{\mathrm{LJ}}(r)$$

and $f(x) = x + \sum_{l=1}^{3} a_l \sin(w_l x)$ is an oscillatory term. For the unbonded particle, the LJ potential is used, as before.

### D.3    MD17

Table 11 shows in detail the performance of the different models on the MD17 dataset.

### D.4    MD22

Table 12 shows in detail the performance of the different models on the MD22 dataset.

Table 10: Huber NLL for the LinPoly-2 dataset

| LinPoly-2 | $O(n)$ GKSN | EMLP | $\pi\,O(n)$ GKSN | $\pi\,O(n)$ GMLP |
|-----------|-------------|------|------------------|------------------|
| m4/n3     | 10.51       | 8.78 | 8.41             | 7.13             |
|           | 0.17        | 0.13 | 0.27             | 0.08             |
| m10/n3    | 8.30        | 7.50 | 7.26             | 4.73             |
|           | 0.40        | 0.18 | 0.39             | 0.78             |
| m10/n5    | 8.36        | 7.77 | 7.18             | 4.11             |
|           | 0.45        | 0.16 | 0.48             | 0.64             |
| m15/n3    | 7.40        | 7.47 | 6.95             | 2.98             |
|           | 0.42        | 0.32 | 0.48             | 0.99             |
| m15/n5    | 7.45        | 7.54 | 6.94             | 2.54             |
|           | 0.45        | 0.17 | 0.56             | 1.00             |

# E    MODEL PARAMETERS

## E.1    HYPER-PARAMETERS AND HYPER-PARAMETER SEARCH

Table 13 show the hyper-parameters used during training for the MLP and GKSN-based architectures. We implemented a separate hyper-parameter search on both MLP and GKSN architecture based on the synthetic dataset, we tested the different sizes of architecture: small (128/16), medium (256/32), and large (512/64); and selected the small for both systems.

While GKSN networks use Spline as the basis, we experimented with ReLU, GeLU, Sigmoid, and Chebichev Polynomial, ReLU provided the most reliable solution across test cases.

## E.2    LJ

Table 14 shows the number of parameters per model for the LJ experiments with $m = 4$ and $n = 3$. The impact of the presentation is already visible. GKSN is always smaller. Table 15 and Table 16 show the network size for $m = 15$ and $n = 3, 5$. As the input increases, the GKSN has more parameters than the equivalent MLP.

## E.3    MD17

Table 17 shows the number of parameters for the models used in the experiments. The permutation invariant version reduces the need for parameters considerably.

## E.4    MD22

As for the MD17 dataset, also for MD22, Table 18 shows the number of parameters for the models used in the experiments. The permutation invariant version reduces the need for parameters considerably.

# F    ABLATION STUDY IN INVARIANTS

Table 19 shows the effect of using different invariant features on the performance in terms of NLL for the Buckyball-catcher system of the MD22 dataset.

We first define some quantities:

$$\|\boldsymbol{x}_i \otimes \boldsymbol{y}_j\| = \sqrt{\|\boldsymbol{x}_i\|^2 \|\boldsymbol{y}_j\|^2 - \langle \boldsymbol{x}_i, \boldsymbol{y}_j \rangle^2}$$

$$\overline{\|\boldsymbol{x}_i \otimes \boldsymbol{y}_j\|} = \|\boldsymbol{x}_i \otimes \boldsymbol{y}_j\| / (\|\boldsymbol{x}_i\| \|\boldsymbol{y}_j\|),$$

$$\overline{\langle \boldsymbol{x}_i, \boldsymbol{y}_j \rangle} = \langle \boldsymbol{x}_i, \boldsymbol{y}_j \rangle / (\|\boldsymbol{x}_i\| \|\boldsymbol{y}_j\|),$$

$$\|\boldsymbol{x}_i \otimes \boldsymbol{y}_j\| = \sqrt{\|\boldsymbol{x}_i\|^2 \|\boldsymbol{y}_j\|^2 - \langle \boldsymbol{x}_i, \boldsymbol{y}_j, \rangle^2}$$

Table 11: Huber NLL for the MD17 dataset

| Dataset (MD17) | aspirin | | benzene2017 | | ethanol | | malonaldehyde | | naphthalene | | salicylic | | toluene | | uracil | |
|---|---|---|---|---|---|---|---|---|---|---|---|---|---|---|---|---|
| $O(n)$ GKSN (F,F) | 6.77 | 0.16 | 8.02 | 0.09 | 7.94 | 0.04 | 7.84 | 0.03 | 7.42 | 0.04 | 7.54 | 0.14 | 7.60 | 0.21 | 8.03 | 0.13 |
| $O(n)$ GKSN (F,T) | 6.08 | 0.01 | 7.29 | 0.03 | 7.14 | 0.03 | 7.12 | 0.04 | 6.29 | 0.08 | 6.41 | 0.03 | 6.50 | 0.06 | 7.08 | 0.00 |
| $O(n)$ GKSN (T,F) | 6.83 | 0.20 | 8.06 | 0.15 | 8.06 | 0.04 | 7.90 | 0.07 | 7.39 | 0.14 | 7.53 | 0.13 | 7.54 | 0.18 | 8.04 | 0.11 |
| $O(n)$ GKSN (T,T) | 6.09 | 0.04 | 7.27 | 0.06 | 7.13 | 0.03 | 7.16 | 0.07 | 6.30 | 0.03 | 6.36 | 0.05 | 6.54 | 0.06 | 7.01 | 0.09 |
| EMLP (F,F) | 5.63 | 0.00 | 5.98 | 0.02 | 5.47 | 0.01 | 5.40 | 0.01 | 5.37 | 0.00 | 5.65 | 0.00 | 5.69 | 0.02 | 5.70 | 0.01 |
| $O(n)$ GMLP (F,T) | 5.63 | 0.01 | 5.91 | 0.00 | 5.46 | 0.03 | 5.42 | 0.02 | 5.34 | 0.00 | 5.62 | 0.01 | 5.71 | 0.00 | 5.61 | 0.00 |
| EMLP (T,F) | 5.61 | 0.01 | 5.93 | 0.02 | 5.41 | 0.01 | 5.37 | 0.01 | 5.36 | 0.01 | 5.61 | 0.00 | 5.64 | 0.04 | 5.69 | 0.01 |
| $O(n)$ GMLP (T,T) | 5.61 | 0.00 | 5.90 | 0.00 | 5.43 | 0.01 | 5.38 | 0.01 | 5.33 | 0.00 | 5.61 | 0.01 | 5.68 | 0.01 | 5.61 | 0.00 |
| $\pi\, O(n)$ GKSN (F,F) | 5.68 | 0.02 | 6.73 | 0.18 | 5.95 | 0.18 | 5.83 | 0.04 | 5.82 | 0.10 | 5.81 | 0.14 | 6.11 | 0.11 | 6.21 | 0.11 |
| $\pi\, O(n)$ GKSN (F,T) | 5.69 | 0.02 | 6.27 | 0.10 | 6.24 | 0.13 | 5.91 | 0.01 | 5.65 | 0.06 | 5.82 | 0.00 | 5.96 | 0.12 | 5.94 | 0.14 |
| $\pi\, O(n)$ GKSN (T,F) | 5.69 | 0.02 | 6.69 | 0.19 | 6.01 | 0.07 | 5.80 | 0.03 | 5.84 | 0.10 | 5.90 | 0.16 | 6.06 | 0.11 | 6.32 | 0.05 |
| $\pi\, O(n)$ GKSN (T,T) | 5.69 | 0.02 | 6.34 | 0.20 | 6.15 | 0.16 | 5.87 | 0.07 | 5.59 | 0.11 | 5.80 | 0.10 | 5.98 | 0.05 | 5.93 | 0.16 |
| $\pi\, O(n)$ GMLP (F,F) | 4.28 | 0.39 | 5.73 | 0.10 | 5.55 | 0.08 | 5.41 | 0.05 | 5.07 | 0.16 | 5.27 | 0.03 | 5.41 | 0.06 | 5.58 | 0.07 |
| $\pi\, O(n)$ GMLP (F,T) | 5.45 | 0.05 | 5.77 | 0.05 | 5.49 | 0.01 | 5.40 | 0.03 | 5.29 | 0.02 | 5.53 | 0.06 | 5.64 | 0.04 | 5.58 | 0.02 |
| $\pi\, O(n)$ GMLP (T,F) | 3.84 | 0.59 | 5.47 | 0.26 | 5.44 | 0.01 | 5.34 | 0.04 | 3.08 | 2.83 | 4.41 | 0.82 | 5.07 | 0.27 | 5.47 | 0.03 |
| $\pi\, O(n)$ GMLP (T,T) | 5.34 | 0.06 | 5.58 | 0.11 | 5.46 | 0.01 | 5.37 | 0.03 | 5.17 | 0.03 | 5.48 | 0.07 | 5.49 | 0.05 | 5.45 | 0.08 |

Table 12: Huber NLL for the MD22 dataset

| Dataset (MD22) | AT-AT-CG-CG | | AT-AT | | Ac-Ala3-NHMe | | DHA | | buckyball-catcher | | stachyose | |
|---|---|---|---|---|---|---|---|---|---|---|---|---|
| $O(n)$ GKSN (F,F) | NaN | NaN | 7.42 | 0.30 | 5.95 | 0.07 | 5.71 | 0.10 | NaN | NaN | NaN | NaN |
| $O(n)$ GKSN (F,T) | 7.94 | 0.19 | 7.27 | 0.18 | 5.65 | 0.03 | 5.59 | 0.01 | 8.92 | 0.20 | 6.24 | 0.11 |
| $O(n)$ GKSN (T,F) | NaN | NaN | 7.20 | 0.33 | 5.85 | 0.13 | 5.67 | 0.14 | NaN | NaN | NaN | NaN |
| $O(n)$ GKSN (T,T) | 8.10 | 0.09 | 7.38 | 0.05 | 5.64 | 0.04 | 5.56 | 0.01 | 8.77 | 0.28 | 6.36 | 0.13 |
| EMLP (F,F) | 7.66 | 0.07 | 6.57 | 0.01 | 5.58 | 0.00 | 5.53 | 0.00 | 7.30 | 0.00 | 5.74 | 0.00 |
| $O(n)$ GMLP (F,T) | 7.64 | 0.03 | 6.57 | 0.01 | 5.58 | 0.00 | 5.51 | 0.00 | 7.27 | 0.00 | 5.70 | 0.01 |
| $O(n)$ EMLP (T,F) | 7.57 | 0.04 | 6.56 | 0.01 | 5.57 | 0.01 | 5.53 | 0.00 | 7.25 | 0.02 | 5.70 | 0.02 |
| $O(n)$ GMLP (T,T) | 7.59 | 0.05 | 6.55 | 0.03 | 5.56 | 0.00 | 5.51 | 0.00 | 7.27 | 0.01 | 5.67 | 0.00 |
| $\pi\,O(n)$ GKSN (F,F) | NaN | NaN | 6.60 | 0.02 | 5.58 | 0.01 | 5.49 | 0.00 | NaN | NaN | NaN | NaN |
| $\pi\,O(n)$ GKSN (F,T) | 7.71 | 0.04 | 6.61 | 0.07 | 5.56 | 0.00 | 5.51 | 0.01 | 7.43 | 0.08 | 5.70 | 0.02 |
| $\pi\,O(n)$ GKSN (T,F) | 7.75 | NaN | 6.61 | 0.02 | 5.57 | 0.00 | 5.50 | 0.01 | NaN | NaN | 5.77 | NaN |
| $\pi\,O(n)$ GKSN (T,T) | 7.74 | 0.07 | 6.64 | 0.03 | 5.56 | 0.01 | 5.52 | 0.01 | 7.39 | 0.06 | 5.73 | 0.03 |
| $\pi\,O(n)$ GMLP (F,F) | NaN | NaN | NaN | NaN | -0.04 | 1.53 | -0.59 | 0.11 | NaN | NaN | NaN | NaN |
| $\pi\,O(n)$ GMLP (F,T) | 1.25 | 0.36 | 2.10 | 0.36 | 3.76 | 0.20 | 2.09 | 0.35 | 0.61 | 1.21 | 1.27 | 1.82 |
| $\pi\,O(n)$ GMLP (T,F) | NaN | NaN | -1.29 | 0.22 | 0.23 | 0.26 | -2.68 | 1.40 | NaN | NaN | NaN | NaN |
| $\pi\,O(n)$ GMLP (T,T) | 0.39 | 0.28 | 1.64 | 0.62 | 1.99 | 2.35 | 1.35 | 1.43 | -0.18 | 0.20 | 1.46 | 1.01 |

| Parameter | Value | Comment |
|---|---|---|
| Number of epochs | 500 | We use 500 for the MD17 and MD22, while 1000 for the LJ experiments |
| batch size | 4092 | |
| loss | Huber | We selected Huber, compared to MSE, since it enables better training |
| em lr | 0.01, | learning rate for energy minimization for LJ experiments |
| em niters | 500 | number of steps for energy minimization for LJ experiments |
| learning rate | 0.001 | we experimented with multiple rate and fix this for all experiments |
| num samples | 10000 | We fix the number of samples, if the dataset contains more data, we first permute the data (same for all experiments) and select the first 10000 samples. |
| trsamples | 8000 | we split 80/20 training and testing |
| optimizer | AdamW | |
| weight decay | $1e-9$ | Weight decay is used to stabilize the training |
| scheduler | ReduceLROnPlateau | The scheduler helps with different system requirement |
| GKSN layers | [input dim, 16, 16, 1] | the architecture size has been selected in the hyper-parameter search |
| GKSN orders | [8,8,8] | This is the number of basis per function |
| GKSN Basis | ReLU | While GKSN networks use Spline as basis, we experimented with ReLU, GeLU, Sigmoid, and Chebichev Polynomial, ReLU provided the most reliable solution |
| MLP layers | [input dim, 128, 128, 1] | the architecture size has been selected in the hyper-parameter search |

Table 13: Hyper-parameters used during training

Table 14: Network sizes during the $4/3$ experiments

| system | model | options | size |
|---|---|---|---|
| m4/n3 | $O(n)$ GKSN | FF | 9911 |
| m4/n3 | $O(n)$ GKSN | FT | 9911 |
| m4/n3 | $O(n)$ GKSN | TF | 12044 |
| m4/n3 | $O(n)$ GKSN | TT | 12044 |
| m4/n3 | EMLP | FF | 22145 |
| m4/n3 | $O(n)$ GMLP | FT | 22145 |
| m4/n3 | $O(n)$ EMLP | TF | 23681 |
| m4/n3 | $O(n)$ GMLP | TT | 23681 |
| m4/n3 | $\pi O(n)$ GKSN | FF | 4167 |
| m4/n3 | $\pi O(n)$ GKSN | FT | 4167 |
| m4/n3 | $\pi O(n)$ GKSN | TF | 4475 |
| m4/n3 | $\pi O(n)$ GKSN | TT | 4475 |
| m4/n3 | $\pi O(n)$ GMLP | FF | 17665 |
| m4/n3 | $\pi O(n)$ GMLP | FT | 17665 |
| m4/n3 | $\pi O(n)$ GMLP | TF | 17921 |
| m4/n3 | $\pi O(n)$ GMLP | TT | 17921 |

Table 15: Network sizes during the $15/3$ experiments

| system | model | options | size |
|--------|-------|---------|------|
| m15/n3 | $O(n)$ GKSN | FF | 250887 |
| m15/n3 | $O(n)$ GKSN | FT | 63691 |
| m15/n3 | $O(n)$ GKSN | TF | 371803 |
| m15/n3 | $O(n)$ GKSN | TT | 87687 |
| m15/n3 | $O(n)$ EMLP | FF | 110849 |
| m15/n3 | $O(n)$ GMLP | FT | 51713 |
| m15/n3 | $O(n)$ EMLP | TF | 137729 |
| m15/n3 | $O(n)$ GMLP | TT | 61697 |
| m15/n3 | $\pi O(n)$ GKSN | FF | 4167 |
| m15/n3 | $\pi O(n)$ GKSN | FT | 4167 |
| m15/n3 | $\pi O(n)$ GKSN | TF | 4475 |
| m15/n3 | $\pi O(n)$ GKSN | TT | 4475 |
| m15/n3 | $\pi O(n)$ GMLP | FF | 17665 |
| m15/n3 | $\pi O(n)$ GMLP | FT | 17665 |
| m15/n3 | $\pi O(n)$ GMLP | TF | 17921 |
| m15/n3 | $\pi O(n)$ GMLP | TT | 17921 |

Table 16: Network sizes during the $15/5$ experiments

| system | model | options | size |
|--------|-------|---------|------|
| m15/n5 | $O(n)$ GKSN | FF | 250887 |
| m15/n5 | $O(n)$ GKSN | FT | 111906 |
| m15/n5 | $O(n)$ GKSN | TF | 371803 |
| m15/n5 | $O(n)$ GKSN | TT | 159216 |
| m15/n5 | $O(n)$ EMLP | FF | 110849 |
| m15/n5 | $O(n)$ GMLP | FT | 70529 |
| m15/n5 | $O(n)$ EMLP | TF | 137729 |
| m15/n5 | $O(n)$ GMLP | TT | 85889 |
| m15/n5 | $\pi O(n)$ GKSN | FF | 4167 |
| m15/n5 | $\pi O(n)$ GKSN | FT | 4167 |
| m15/n5 | $\pi O(n)$ GKSN | TF | 4475 |
| m15/n5 | $\pi O(n)$ GKSN | TT | 4475 |
| m15/n5 | $\pi O(n)$ GMLP | FF | 17665 |
| m15/n5 | $\pi O(n)$ GMLP | FT | 17665 |
| m15/n5 | $\pi O(n)$ GMLP | TF | 17921 |
| m15/n5 | $\pi O(n)$ GMLP | TT | 17921 |

Table 17: Network sizes during the aspirin experiments

| dataset | model | options | size |
|---------|-------|---------|------|
| aspirin | $O(n)$ GKSN | FF | 1186625 |
| aspirin | $O(n)$ GKSN | FT | 147811 |
| aspirin | $O(n)$ GKSN | TF | 1692200 |
| aspirin | $O(n)$ GKSN | TT | 197535 |
| aspirin | $O(n)$ EMLP | FF | 258689 |
| aspirin | $O(n)$ GMLP | FT | 82433 |
| aspirin | $O(n)$ EMLP | TF | 312449 |
| aspirin | $O(n)$ GMLP | TT | 97025 |
| aspirin | $\pi O(n)$ GKSN | FF | 4475 |
| aspirin | $\pi O(n)$ GKSN | FT | 4475 |
| aspirin | $\pi O(n)$ GKSN | TF | 4783 |
| aspirin | $\pi O(n)$ GKSN | TT | 4783 |
| aspirin | $\pi O(n)$ GMLP | FF | 17921 |
| aspirin | $\pi O(n)$ GMLP | FT | 17921 |
| aspirin | $\pi O(n)$ GMLP | TF | 18177 |
| aspirin | $\pi O(n)$ GMLP | TT | 18177 |

Table 18: Network sizes during the AT-AT-CG-CG experiments

| dataset | model | options | size |
|---------|-------|---------|------|
| AT-AT-CG-CG | $O(n)$ GKSN | FF | 974480535 |
| AT-AT-CG-CG | $O(n)$ GKSN | FT | 2938488 |
| AT-AT-CG-CG | $O(n)$ GKSN | TF | 1453151821 |
| AT-AT-CG-CG | $O(n)$ GKSN | TT | 4256886 |
| AT-AT-CG-CG | $O(n)$ EMLP | FF | 7969025 |
| AT-AT-CG-CG | $O(n)$ GMLP | FT | 417665 |
| AT-AT-CG-CG | $O(n)$ EMLP | TF | 9736193 |
| AT-AT-CG-CG | $O(n)$ GMLP | TT | 506753 |
| AT-AT-CG-CG | $\pi O(n)$ GKSN | FT | 4475 |
| AT-AT-CG-CG | $\pi O(n)$ GKSN | TF | 4783 |
| AT-AT-CG-CG | $\pi O(n)$ GKSN | TT | 4783 |
| AT-AT-CG-CG | $\pi O(n)$ GMLP | FF | 17921 |
| AT-AT-CG-CG | $\pi O(n)$ GMLP | FT | 17921 |
| AT-AT-CG-CG | $\pi O(n)$ GMLP | TF | 18177 |
| AT-AT-CG-CG | $\pi O(n)$ GMLP | TT | 18177 |

Table 19: Hubert NLL or the Buckyball-catcher system of the MD22 dataset, with the linear version of the representation and with the node id for the $\pi O(n)$ GKSN model.

| Features | Train NLL | | Test NLL | |
|----------|-----------|---|----------|---|
| cos | 6.80 | $\pm 0.12$ | 6.49 | $\pm 0.03$ |
| sin-cos | 6.67 | $\pm 0.08$ | 5.97 | $\pm 0.62$ |
| n1 | 6.77 | $\pm 0.07$ | 6.63 | $\pm 0.05$ |
| n1-n12 | 6.70 | $\pm 0.16$ | 6.48 | $\pm 0.20$ |
| n12 | 6.65 | $\pm 0.02$ | 6.47 | $\pm 0.11$ |
| inner | 6.69 | $\pm 0.00$ | 4.69 | $\pm 2.50$ |
| inner-n1 | 6.79 | $\pm 0.01$ | 6.64 | $\pm 0.09$ |
| inner-n1-n12 | 6.65 | $\pm 0.25$ | 6.45 | $\pm 0.20$ |
| inner-outer | 6.82 | $\pm 0.03$ | **6.67** | $\pm 0.00$ |
| inner-outer-n1 | 6.58 | $\pm 0.21$ | 6.48 | $\pm 0.27$ |
| inner-outer-n1-n12 | 6.82 | $\pm 0.01$ | **6.67** | $\pm 0.02$ |

Table 20: More detailed ablation study, showing the Hubert NLL synthetic dataset $m = 5, n = 2$.

| method | feature | Node Id | Linear | train | std | test | std |
|---|---|---|---|---|---|---|---|
| $O(n)$ GKSN | all | False | False | 6.20 | 0.30 | 6.15 | 0.37 |
| | | | True | 6.20 | 0.29 | 6.15 | 0.37 |
| | inner-outer | False | False | 6.20 | 0.30 | 6.16 | 0.37 |
| | | | True | 6.20 | 0.30 | 6.16 | 0.37 |
| | n1 | False | False | 6.19 | 0.30 | 6.17 | 0.37 |
| | | | True | 6.18 | 0.30 | 6.17 | 0.37 |
| | n12 | False | False | 6.19 | 0.30 | 6.17 | 0.37 |
| | | | True | 6.18 | 0.30 | 6.17 | 0.37 |
| | sin-cos | False | False | 5.90 | 0.31 | 5.91 | 0.41 |
| | | | True | 5.91 | 0.30 | 5.91 | 0.41 |
| $\pi O(n)$ MLP | all | False | False | 6.23 | 0.41 | 6.22 | 0.53 |
| | | True | False | 6.23 | 0.40 | 6.22 | 0.52 |
| | inner-outer | False | False | 6.16 | 0.34 | 6.10 | 0.48 |
| | | True | False | 6.19 | 0.30 | 6.17 | 0.37 |
| | n1 | False | False | 6.19 | 0.29 | 6.17 | 0.37 |
| | | True | False | 6.23 | 0.40 | 6.21 | 0.50 |
| | n12 | False | False | 6.14 | 0.36 | 6.15 | 0.40 |
| | | True | False | 6.21 | 0.42 | 6.09 | 0.69 |
| | sin-cos | False | False | 5.92 | 0.30 | 5.91 | 0.41 |
| | | True | False | 5.92 | 0.30 | 5.91 | 0.41 |
| $O(n)$ EMLP | all | False | False | 6.20 | 0.29 | 6.17 | 0.38 |
| | | True | False | 6.20 | 0.29 | 6.18 | 0.38 |
| | inner-outer | False | False | 6.16 | 0.34 | 6.10 | 0.48 |
| | | True | False | 6.19 | 0.30 | 6.17 | 0.37 |
| | n1 | False | False | 6.19 | 0.29 | 6.17 | 0.37 |
| | | True | False | 6.19 | 0.29 | 6.17 | 0.36 |
| | n12 | False | False | 6.14 | 0.36 | 6.15 | 0.40 |
| | | True | False | 6.18 | 0.30 | 6.09 | 0.49 |
| | sin-cos | False | False | 5.92 | 0.30 | 5.91 | 0.41 |
| | | True | False | 5.92 | 0.30 | 5.91 | 0.41 |

Table 21: Hubert NLL for the MD22 dataset for $\pi O(n)$ GKSN

| Num. samples | Train NLL | Test NLL |
|---|---|---|
| 100 | $6.78\ ^{1.15}$ | $3.01\ ^{0.31}$ |
| 500 | $4.90\ ^{0.08}$ | $4.40\ ^{0.04}$ |
| 1000 | $4.96\ ^{0.11}$ | $4.68\ ^{0.10}$ |
| 3000 | $6.09\ ^{0.11}$ | $5.95\ ^{0.11}$ |
| 6102 | $6.86\ ^{0.09}$ | $6.74\ ^{0.10}$ |

We can now define the features used as input to the representation, which are:

$$\text{n1: } \|\boldsymbol{x}_i\|, \|\boldsymbol{y}_j\|,$$
$$\text{n12: } \|\boldsymbol{x}_i - \boldsymbol{y}_j\|,$$
$$\text{inner: } \langle \boldsymbol{x}_i, \boldsymbol{y}_j \rangle,$$
$$\text{outer: } \|\boldsymbol{x}_i \otimes \boldsymbol{y}_j\|$$
$$\text{cos: } \overline{\langle \boldsymbol{x}_i, \boldsymbol{y}_j \rangle},$$
$$\text{sin: } \overline{\|\boldsymbol{x}_i \otimes \boldsymbol{y}_j\|}$$

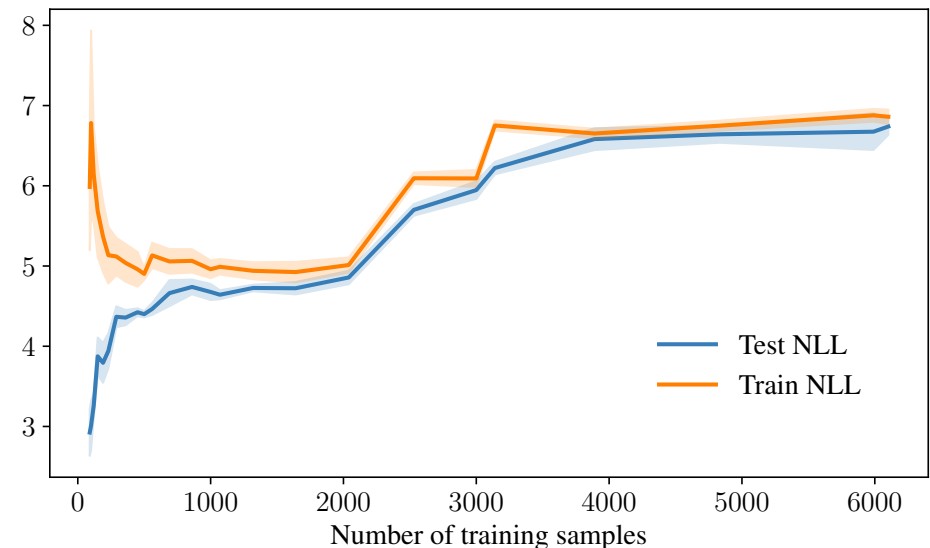

Figure 6: The train and test accuracy in terms of NLL for the Buckyball-catcher system of the MD22 dataset with invariant features inner-outer-n1-n12.

## G  EFFICIENCY OF TRAINING: GROUP INVARIANCE AND NUMBER OF SAMPLES

In Figure 6 and Table 21, we show the effect of the sample size for training the invariant representation. In order to obtain a high test accuracy, more than 50% of the data is necessary.

## H  NUMBER OF PARAMETERS

The following table shows the number of parameters for the different configurations. They showcase the advantage of the introduced representation (numbers are also reported in Annex E). With the selected architecture (see hyperparameter in Section E.1 / Table 13) the number of parameters of GKSN is larger than MLP for the quadratic version, but only slightly higher for the linear representation, which itself is lower than the quadratic representation. This suggests that the representations introduced by our paper have practical implications in reducing the number of parameters for both the GKSN and MLP. We particularly notice that the reduction in the number of parameters from the MLP to the GSKN version for the $S(m)+O(n)$ architecture is 75% with the increase in the $O(n)$ being of .1% for the linear representation (see m10/n3 case, m is the number of nodes and n is the dimension of the euclidean space, including the Id of the node is important for the $S(m)$ representation to distinguish different nodes).

## I  TRAINING TIMES

The following table shows the difference in training time for the different representations. Please note that $O(n)$-MLP (Quadratic), corresponds to EMLP proposed in (Villar et al., 2021). We observe that there is an increase in computation cost from MLP to GKSN of about 10% for the O(n) architecture, while it is between 20% and 30% for the permutation invariant S(m) architecture. Further, we note that the use of $S(m)$ representation introduces a penalty of between 1% and 6% for the MLP case. The current GKSN implementation is not optimized for speed, which may influence the actual values.

In conclusion, we show that the introduction of GKSN only introduces a marginal increase in computation cost (10%-30%) while providing a consistent reduction in the parameters count ( 75% for the LJ experiments).

| | Model | Without Id | | With Id | |
|---|---|---|---|---|---|
| | | Quadratic | Linear | Quadratic | Linear |
| m10/n3 | $O(n)$-GKSN | 76,167 | 35,991 | 106,393 | 48,003 |
| | EMLP | 57,089 | 38,273 | 68,609 | 44,417 |
| | $S(m)$-$O(n)$-GKSN | 4,167 | 4,167 | 4,475 | 4,475 |
| | $S(m)O(n)$-MLP | 17,665 | 17,665 | 17,921 | 17,921 |
| m10/n5 | $O(n)$-GKSN | 76,167 | 55,753 | 106,393 | 76,167 |
| | EMLP | 57,089 | 48,129 | 68,609 | 57,089 |
| | $S(m)$-$O(n)$-GKSN | 4,167 | 4,167 | 4,475 | 4,475 |
| | $S(m)$-$O(n)$-MLP | 17,665 | 17,665 | 17,921 | 17,921 |
| m15/n3 | $O(n)$-GKSN | 250,887 | 63,691 | 371,803 | 87,687 |
| | EMLP | 110,849 | 51,713 | 137,729 | 61,697 |
| | $S(m)$-$O(n)$-GKSN | 4,167 | 4,167 | 4,475 | 4,475 |
| | $S(m)$-$O(n)$-MLP | 17,665 | 17,665 | 17,921 | 17,921 |
| m15/n5 | $O(n)$-GKSN | 250,887 | 111,906 | 371,803 | 159,216 |
| | EMLP | 110,849 | 70,529 | 137,729 | 85,889 |
| | $S(m)$-$O(n)$-GKSN | 4,167 | 4,167 | 4,475 | 4,475 |
| | $S(m)$-$O(n)$-MLP | 17,665 | 17,665 | 17,921 | 17,921 |
| m4/n3 | $O(n)$-GKSN | 9,911 | 9,911 | 12,044 | 12,044 |
| | EMLP | 22,145 | 22,145 | 23,681 | 23,681 |
| | $S(m)$-$O(n)$-GKSN | 4,167 | 4,167 | 4,475 | 4,475 |
| | $S(m)$-$O(n)$-MLP | 17,665 | 17,665 | 17,921 | 17,921 |

Table 22: Number of parameters the different architectures of GKSN for the LJ experiments with dimensions m10/n3.

| | m10/n3 | m10/n5 | m15/n3 | m15/n5 | m4/n3 |
|---|---|---|---|---|---|
| $O(n)$-GKSN vs EMLP | | | | | |
| **Without id** | | | | | |
| Quadratic | 9.9% | 8.6% | 11.6% | 11.7% | 10.0% |
| Linear | 9.7% | 8.9% | 13.5% | 7.5% | 12.1% |
| **With Id** | | | | | |
| Quadratic | 14.0% | 17.8% | 12.2% | 11.1% | 9.0% |
| Linear | 10.2% | 8.5% | 10.0% | 15.1% | 6.7% |
| $S(m)$-$O(n)$-GKSN vs $S(m)$-$O(n)$-MLP | | | | | |
| **Without id** | | | | | |
| Quadratic | 39.6% | 26.9% | 13.7% | 2.7% | 14.8% |
| Linear | 30.8% | 20.6% | 41.5% | 31.8% | 28.9% |
| **With Id** | | | | | |
| Quadratic | 19.8% | 23.1% | 12.3% | 4.7% | 13.8% |
| Linear | 28.1% | 15.8% | 24.9% | 27.9% | 25.8% |
| $S(m)$-$O(n)$-GKSN vs EMLP | | | | | |
| **Without id** | | | | | |
| Quadratic | 2.9% | 3.5% | 7.3% | 14.8% | 0.5% |
| Linear | 4.2% | 2.8% | -1.5% | -2.2% | 8.1% |
| **With Id** | | | | | |
| Quadratic | 13.2% | 6.4% | 7.1% | 10.1% | 1.5% |
| Linear | -2.2% | -2.0% | 5.3% | 2.5% | 2.8% |

Table 23: Training times of different architectures of GKSN for the LJ experiments with various dimensions, compared with the equivalent architecture using an MLP representation.

## J ADDITIONAL RELATED WORK

**Symmetry preserving machine learning architecture**   Machine learning interatomic potentials (MLIPs) have emerged as powerful tools for modeling interatomic interactions in molecular and materials systems, offering a computationally efficient alternative to traditional ab initio methods. Architectures like Schnet (Schütt et al., 2017) use continuous-filter convolutional layers to capture local atomic environments and message passing, enabling accurate predictions of molecular properties. To further enhance physical expressivity, $E(3)$-equivariant architectures (Thomas et al., 2018b) have been developed, which respect the symmetries of Euclidean space (rotations, translations, and reflections) by design. These models, such as Tensor Field Networks (Thomas et al., 2018b) and NequIP (Batzner et al., 2022), ensure that predictions (i.e. energy and forces) are invariant or equivariant to transformations in 3D space, making them highly data-efficient for tasks like force field prediction in molecular dynamics. MACE (Batatia et al., 2023) is a higher-order equivariant message-passing network that enhances force field accuracy and efficiency by leveraging multi-body interactions. E(n)-equivariant GNNs (EGNNs) (Satorras et al., 2022) implement a higher-order representation while maintaining equivariance to rotations, translations, and permutations. Irreducible Cartesian Tensor Potential (ICTP) (Zaverkin et al., 2024) introduces irreducible Cartesian tensors for equivariant message passing, offering computational advantages over spherical harmonics in the small tensor rank regime. Tensor field networks (Thomas et al., 2018a) and Equiformer (Liao & Smidt, 2023) use spherical harmonics as bases for tensors. While SO3krates (Frank et al., 2024) combines sparse equivariant representations with transformers to balance accuracy and speed. Additionally, equivariant Clifford networks (Ruhe et al., 2023) extend this framework by incorporating geometric algebra to build equivariant models. Equivariant representations mitigate cumulative errors in molecular dynamics (Unke et al., 2021), while directional message passing with spherical harmonics improves angular dependency modeling as implemented in DimeNet (Gasteiger et al., 2022). Equivariant or invariant architectures enhance data efficiency, accuracy, and physical consistency in tasks where input symmetries (e.g., rotation, reflection, translation) dictate output invariance or equivariance. In collider physics, jet-tagging is the problem of identifying the type of particles that have generated the particle collision jet. The collision jet exhibits space-time symmetry, the Lorentz boost. Symmetry-preserving architecture for the Lorentz group have been proposed architecture based on high-order tensor products as LoLa (Butter et al., 2018), LBN (Erdmann et al., 2019) LGN (Bogatskiy et al., 2020), and LorentzNet (Gong et al., 2022), which introduce Minkowski dot product attention. Finally, permutation preserving models have been proposed to model function over sets, as DeepSet and subsequent models (Zaheer et al., 2017; Amir et al., 2023). While these advancements have significantly improved the accuracy and efficiency of MLIPs for applications in chemistry, physics, and materials science, the advantage of KAN architecture has not yet been explored, we thus take a fundamental step in this direction with our study.

**KAN Architectures**   Kolmogorov-Arnold Networks (KANs) are inspired by the Kolmogorov-Arnold representation theorem, which provides a theoretical foundation for approximating multivariate functions using univariate functions and addition. Early work by Hecht-Nielsen (1987) (Hecht-Nielsen, 1987) introduced one of the first neural network architectures based on this theorem, demonstrating its potential for efficient function approximation. (Lai & Shen, 2021) study the approximation capability of KST-based models in high dimensions and how they could potentially break the curse of dimensionality (Poggio, 2022). (Ferdaus et al., 2024) propose to combine Convolutional Neural Networks (CNNs) with Kolmogorov Arnold Network (KAN) principles. Additionally, (Yang & Wang, 2025) explored the integration of KAN principles into transformer models, achieving improvements in efficiency for sequence modeling tasks. (Hu et al., 2025) propose EKAN, an approximation method for incorporating matrix group equivariance into KANs. While these studies highlight the versatility of KAN architectures in adapting to various neural network frameworks, the extension to physical and geometrical symmetries has not been fully considered.

**Application of KAN**   KANs have been applied to a range of machine learning tasks, particularly in scenarios requiring efficient function approximation. For instance, Kůrková (1991) (Kůrková, 1992) demonstrated the effectiveness of KANs in high-dimensional regression problems, where traditional neural networks often struggle with scalability. In the natural language processing domain, (Galitsky, 2024) utilized KAN for word-level explanations. Furthermore, (Carlo et al., 2024) applied KANs to graph-based learning tasks, showing that their hybrid models could achieve state-of-the-art results in

graph classification and node prediction. KAN has been used as a function approximation to solve PDE (Wang et al., 2024; Shukla et al., 2024) for both forward and backward problems with highly complex boundary and initial conditions. (Aghaei, 2024) extends KAN with rational polynomials basis to regression and classifications problems. (Seydi et al., 2024) explores using Wavelet as basis functions to model hyper-spectral data. KANs have been extended to model time-series (Xu et al., 2024c; Inzirillo & Genet, 2024) to dynamically adapt to temporal data. While these, and other (Somvanshi et al., 2024), applications highlight the practical utility of KANs in solving complex real-world problems, a significant class of molecular applications remains overlooked.

## K  EXPERIMENTAL CODE

Code is provided in anonymized format at the following URL `https://anonymous.4open.science/r/GKSN-37BD/`.

## L  EXISTING ASSETS

We based our experiments on python and Pytorch, here the licenses:

**PyTorch** , PyTorch is released under the Modified BSD License,

**python** "All Python releases are Open Source (see http://www.opensource.org for the Open Source Definition). Historically, most, but not all, Python releases have also been GPL-compatible;"

**numpy** : NumPy is distributed under a BSD license,

**matplotlib** : `https://github.com/matplotlib/matplotlib/blob/main/LICENSE/LICENSE`

## M  USAGE OF LARGE LANGUAGE MODELS

LLM has been used as a tool for writing at the sentence level or as an alternative internet search tool.