# OpenReview forum: "Geometric Kolmogorov Superposition Representation of group invariant function for computational science"
_ICLR.cc/2026/Conference — Submitted to ICLR 2026_

### Official Review · Reviewer_U3N3 · 2025-10-26

**Soundness:** 2
**Presentation:** 3
**Contribution:** 2
**Rating:** 4
**Confidence:** 1

**Summary:**

This paper proposes a variation of Kolmogorov Arnold networks for group invariant tasks such as molecular dynamics simulation.

**Strengths:**

The paper's presentation looks visually clear. I am not well positioned to judge the originality and significance of the work, as I am not directly working on KANs and not sufficiently familiar with all of the maths here.

**Weaknesses:**

Experiments on MD17 and MD22 are insufficient to make claims about molecular simulation capabilities, as these datasets are not used anymore in cutting edge ML Interatomic Potentials (MLIP) papers anymore to the best of my knowledge. For instance, a major limitation is that evaluation on MD17/22 requires training separate models for each molecular system to be simulated. Hence, models are never truly evaluated for generalization to new molecules unseen in training.

Despite the limits of the MD17/22 datasets, the authors have also not compared to any state-of-the-art MLIPs in the tables.

**Questions:**

Following up on the weakness identified regarding the MD17/22 evaluation, I have a question: Is the proposed GKSR architecture able to handle variable sized input molecules?

The authors end the paper with a sentence that I felt was pretty mysterious: "further investigation will show if this architecture can be extended to implement machine learning interatomic potentials". Can you clarify/elaborate this sentence?

What is currently missing from GKSR that prevents it from implementing practical MLIPs?

An other major consideration for MLIPs is speed/runtime: while in theory KANs may help improve MLIPs, how does this impact practical usage speed?

---

> ### Author Response · Authors · 2025-11-24
> **Questions**
>
> We thank the reviewer for the various questions, and we hope we address those in full.
>
>
> > Experiments on MD17 and MD22 are insufficient to make claims about molecular simulation capabilities, as these datasets are not used anymore in cutting edge ML Interatomic Potentials (MLIP) papers anymore to the best of my knowledge.
>
> While the design of an improved architecture is an appealing and worthy task, our paper focuses on an exact representation of invariant functions using KST-based architectures. We then evaluate the learning ability of an implementation of the representation for molecular energy prediction. Since we did not aim at designing the best MLIP, we also did not include a direct comparison with the recent work in this line of work. Nevertheless, also following the request of reviewer bggU, we now compare GKSN against Schnet.
>
>
> >For instance, a major limitation is that evaluation on MD17/22 requires training separate models for each molecular system to be simulated. Hence, models are never truly evaluated for generalization to new molecules unseen in training.
>
> The MD17/22 tasks aim at training and evaluating on a single atomistic system. In order to allow the proper transferability of the trained potential, proper architecture choices have to be introduced. A typical approach in MLIP is to encode 1) the atom type and 2) the local environment within a cutoff radius. This is performed in the additional experiments.
>
>
> > Despite the limits of the MD17/22 datasets, the authors have also not compared to any state-of-the-art MLIPs in the tables.
>
> We have extended the experiment by comparing it to the Schnet architecture. See the reviewer bggU’s response. We, however, would like to emphasize that our focus is to explore the exact representation of invariant functions using the KST-based representation, and the representation is not specific to a particular task, such as the energy and force prediction for an atomic system.
>
>
> >Following up on the weakness identified regarding the MD17/22 evaluation, I have a question: Is the proposed GKSR architecture able to handle variable sized input molecules?
>
> The basic architecture is not, since it has not been designed for this task; we reported the experiments against the Schnet architecture, which includes these biases.
>
>
> >The authors end the paper with a sentence that I felt was pretty mysterious: "further investigation will show if this architecture can be extended to implement machine learning interatomic potentials". Can you clarify/elaborate this sentence?
>
> Our paper explores the exact representation of invariant functions using the KST-based representation; therefore, it is not specific to a particular task, such as energy and force prediction for an atomic system. On the other hand, it is possible to design an architecture that uses our representation for these tasks. This is also shown with the additional experiments against Schnet.
>
>
> > What is currently missing from GKSR that prevents it from implementing practical MLIPs?
>
> MLIP architecture includes different components that would need to be introduced, as for example: 1) local representation of the atomistic system within the cutoff radius, 2) some mechanism to propagate this information 3) possibly constructing some multi-body features used to propagate information among local environments.
>
>
> >An other major consideration for MLIPs is speed/runtime: while in theory KANs may help improve MLIPs, how does this impact practical usage speed?
>
> We observe indeed an increase in computational complexity, mainly induced by the temporary expansion of the tensor size during the computation of the representation. At the same time, we observe a reduction in the number of parameters.
>
>
> We hope we address the major concerns of the reviewer and that the reviewer will reconsider the score.

---

### Official Review · Reviewer_7yyd · 2025-10-31

**Soundness:** 3
**Presentation:** 3
**Contribution:** 3
**Rating:** 6
**Confidence:** 3

**Summary:**

The paper introduces Geometric Kolmogorov Superposition Representation (GKSR) and a practical GKSN layer that extend KST/KAN to explicitly handle group symmetries (O(n), O(1, n), GL(n), and permutations). The core idea is to express invariant functions via univariate functions applied to group invariants (e.g., inner products), with a tightened representation that reduces feature growth from quadratic in nodes to linear by projecting onto a spanning set. The network implementation adds a near-linear “residual-like” path to ease optimization. Empirically, GKSN matches or outperforms strong equivariant/physics baselines on Lennard-Jones, MD17/MD22, and Lorentz-invariant jet-tagging tasks.

**Strengths:**

- The paper is technically sound and (to my knowledge) appears to be the first KAN-family work that systematically incorporates group symmetries within a clean formulation.
- The presentation is rigorous, well organized, and self-consistent.
- The proposed GKSN shows competitive empirical performance, standing as a solid proof of concept supported by comprehensive experiments, including ablation studies (e.g., with and without heuristic invariants) and compelling applications such as symmetry discovery.

**Weaknesses:**

- I find no critical weaknesses. A minor limitation is the inclusion of certain heuristic, handcrafted invariants (e.g., norms or pairwise distances, …), which slightly compromise the purity of the theoretical framework. While this is understandable from an implementation standpoint, it blurs the line between learned and prescribed invariants.

- I tried to review the code, but the anonymous GitHub repository returns ‘The requested file is not found’ for all files.

**Questions:**

This is not a criticism but a conceptual curiosity:
- Can the invariant construction be made learnable (e.g., by parameterizing the bilinear form) to adapt to approximate or broken symmetries?

---

> ### Author Response · Authors · 2025-11-24
> **Clarifications**
>
> We thank the reviewer for the kind words and for acknowledging our contribution. We address the remaining concerns.
>
> > I find no critical weaknesses. A minor limitation is the inclusion of certain heuristic, handcrafted invariants
>
> Thank you for the comment. The theoretical result is not affected. While we have to leave this particular task open for future improvement, we want to note that our work is mostly focused on building an exact representation of a large class of group actions (i.e. symmetries) for invariant functions to these groups.
>
> We acknowledge that when designing the learning method, architectural decisions need to be introduced, as for example extending the invariant features. We, for example, explore this direction in order to reduce the need to learn basic transformations of invariant features. While we agree this may diverge from the original intent of universality, we wanted to show the influence of this factor, and others, in the learning ability.
>
> We therefore leave for future work the proper engineering of the most appropriate architecture that takes advantage of the structure we introduce. We hope this resolves the concern of the reviewer.
>
> > I tried to review the code, but the anonymous GitHub repository returns ‘The requested file is not found’ for all files.
>
> The code share is minimal, and we will release the full code once accepted, after proper cleaning and including additional experiments. We use the anonymous provider that may not always be responsive; indeed also for us the website was not always available. We apologize for the inconvenience.

---

### Official Review · Reviewer_bggU · 2025-11-01

**Soundness:** 2
**Presentation:** 2
**Contribution:** 2
**Rating:** 4
**Confidence:** 2

**Summary:**

In this submission, the authors propose the Geometric Kolmogorov Superposition Representation (GKSN), which extends the Kolmogorov–Arnold theorem (KAT) to settings where functions are constrained by symmetry groups such as $E(3), O(1), O(1, n), S_n$, and $GL$. Motivated by this theoretical formulation, the proposed architectures are implemented and evaluated on the MD17 and MD22 datasets.

**Strengths:**

1. This work represents a meaningful step toward incorporating symmetry constraints into the KAN family of models, as symmetry is a fundamental property in physical systems.

2. The work considers widely discussed symmetries into the framework including  $E(3), O(1), O(1, n), S_n$, and $GL$.

**Weaknesses:**

1. Experiments on MD17/MD22. These benchmarks primarily evaluate atomistic energy (and often forces). Please report the absolute energy error (e.g., MAE in meV) and, if applicable, force error (meV/Å). It would be helpful to include efficiency metrics—such as parameter count, FLOPs, and wall-clock time per epoch—and a direct comparison with invariant GNN baselines (e.g., SchNet) to better understand GKSN’s accuracy–efficiency trade-off.

2. Positioning vs. existing invariant GNNs. The comparison and claimed advantages over established invariant architectures remain unclear. What is the computational complexity of GKSN relative to standard invariant GNNs, and how does its performance scale with model size?

3. Relation to frame averaging. Many invariant GNNs handle multiple symmetries through frame averaging. Please elaborate on how GKSN differs conceptually and practically from such approaches. What are the trade-offs in terms of inductive bias, sample efficiency, and computational cost? Under what conditions would GKSN be expected to outperform frame-averaged architectures?

4. Overall, the idea is promising and potentially impactful, but the work would benefit from clearer quantitative comparisons and a deeper discussion of how GKSN advances upon and relates to existing invariant GNN methods and frame-averaging techniques.

**Questions:**

N/A.

---

> ### Author Response · Authors · 2025-11-24
> **Experiments**
>
> > ”Experiments  [...] for atomistic energy and forces, [...] parameter count,  [...] comparison”
>
> The experiments on MD17/22 have been designed to study the behaviour for learning an invariant function to $O(3)$ actions on a realistic dataset. Table 23 provides the evaluation of the computation time difference.
> We thank the reviewer for the request to extend the comparison to other equivariant architectures. We therefore explore the performance with respect to Schnet, where we apply GKSN.  To this end, we used Schnet implemented in the Pytorch-Geometric library that also provides the MD17 dataset.
> We compare both total energy and force predictions, where the forces are computed as $\nabla_x E_\theta(x)$, with $E_\theta(x)$ the energy of the atomic system.
> We present the computation time per epoch, numbers of parameters (selected to be ~ equal), and the MAE for both forces and energies for the train, and test splits. The table shows MAEs of the total energy (in kcal/mol) and forces (kcal/mol/Angstrom) per atom.
> Experimental set-up: 200 epochs, 30k training, 5k testing, batch_size 32, lr 0.001, linear scheduling of the weight of the energy, for 1 to 1/40.
> Common params: cutoff  = 10.0, num_interactions = 3 , hidden_channels  = 32 , num_gaussians = 20
> GKSN: nr. parameters: 33’384, training 39.1s/epoch, test 3.9s, num_filters=24
> SCHNET: nr. parameters: 35’809, training 25.3s/epoch, test 2.5s, num_filters=64
>
> | Dataset | Model | $\text{MAE}_\text{energy}^{tr}$ | $\text{MAE}_\text{energy}^{te}$ | $\text{MAE}_\text{force}^{tr}$ | $\text{MAE}_\text{force}^{te}$ |
> | :---  |  :---  | :--- | :---  |  :---  | :--- |
> | benzene | GKSN | ${30}^{5}$ | ${289}^{172}$ | ${8}^{3}$ | ${6}^{2}$ |
> | | SCHNET | ${4469}^{1604}$ | ${672}^{291}$ | ${15}^{6}$ | ${1}^{0}$ |
> | ethanol | GKSN | ${28}^{7}$ | ${163}^{53}$ | ${7}^{1}$ | ${3}^{0}$ |
> | | SCHNET | ${2304}^{1000}$ | ${318}^{174}$ | ${16}^{7}$ | ${2}^{1}$ |
> | toluene | GKSN | ${60}^{9}$ | ${361}^{121}$ | ${9}^{1}$ | ${6}^{1}$ |
> | | SCHNET | ${3860}^{2215}$ | ${245}^{251}$ | ${17}^{6}$ | ${2}^{1}$ |
> | uracil | GKSN | ${63}^{17}$ | ${411}^{265}$ | ${13}^{3}$ | ${6}^{4}$ |
> | | SCHNET | ${7586}^{2066}$ | ${541}^{520}$ | ${33}^{9}$ | ${3}^{0}$ |
>
> > Positioning vs. existing invariant GNNs
>
> Table 1, from the theory, shows the number of functions needed. Approximations are necessary when we do not have access to the univariate functions. Specialized architectures are used for specific applications.
> The main theoretical results tell us that the number of univariate functions we need is the same as the non-invariant version.
> The non-invariant version can represent an invariant function only approximately. The invariant version typically requires substantially less data. We will add some clarifications in the manuscript.
>
> > Relation to frame averaging
>
> Frame averaging (FA) requires applying the group actions to a function to transform it into an invariant or equivariant function. In its basic form, its complexity is proportional to the number of elements of the group; therefore, for rotation or in general, a continuous group, the complexity is, in theory, infinite. [Puny et al.] propose to reduce the number of group actions using a frame $F(X)$, which contains a “carefully selected subset of G”, and propose to sample from the F(X) set to reduce complexity. However, while the expressive power of FA is also shown in Theorem 4, together with the error bound of the FA version of the underlying function, the representation of our paper is exact. The FA, on the other hand, maintains the expressive power of the underlying function and therefore is able to use powerful architectures. For the general case, the authors propose to sample actions, thus introducing approximation, while for the Euclidean case, the author proposes to use PCA (principal component analysis) to extract key axes of the system and define a frame of dimension $2^d$, which therefore increases the complexity of the underly model of a factor exponential with the number of dimensions. We will add this analysis to the related work session.
>
> > “GKSN advances upon and relates to existing invariant GNN methods and frame-averaging techniques.”
>
> We thank the reviewer for the comment. The framework we propose in our work is general and shows an alternative representation of invariant functions. Our paper explores the exact representation of invariant functions using KST-based architectures, while, for example, FA, aims at providing a framework to enable existing architectures to approximate invariant and equivariant functions. We then analyze the practical learning ability of approximate architectures. With the additional experiments and discussions, we hope we have addressed the concerns of the relationship with other invariant approaches.
>
> We hope we addressed the Reviewer’s concerns. The Reviewer could consider updating the score in light of our responses. We remain available for any further clarification.

---

### Official Review · Reviewer_wwYp · 2025-11-01

**Soundness:** 3
**Presentation:** 3
**Contribution:** 3
**Rating:** 6
**Confidence:** 3

**Summary:**

This paper introduces the Geometric Kolmogorov Superposition Representation (GKSR), a novel theoretical framework that extends the Kolmogorov-Arnold Theorem (KAT) to inherently incorporate geometric symmetries. The primary goal is to address a key limitation of standard Kolmogorov-Arnold Networks (KANs), which struggle to model physical systems requiring $E(3)$ invariance or other group symmetries.

**Strengths:**

1. The paper's primary strength is its originality. It provides the first, to my knowledge, principled theoretical framework for unifying the exact representation property of KAT with the geometric consistency required in scientific computing.

2. The experiments, particularly on the MD17 and MD22 datasets, show a clear and consistent performance benefit for GKSN over the equivalent EMLP/GMLP baselines.

**Weaknesses:**

1. The paper's key theoretical improvement reduces complexity from a naive $O(m^4)$ to $O(m^2n^2)$. While a significant theoretical step, this is still quadratic in the number of particles ($m$).

2. The authors motivate GKSN using the exact representation property of KAT . However, the practical implementation (GKSN) uses ReLU as the basis for the univariate functions, not the splines or other smooth functions typically required for the theoretical guarantee. This choice is likely for practical efficiency, but it weakens the connection to the core theoretical motivation.

**Questions:**

1. Could the authors comment on the expected performance of a naive KAN (without the invariant-feature preprocessing) on these tasks? This baseline seems critical to understanding whether the performance gains stem from the KAN architecture itself or from the new geometric representation.

2. The empirical results (e.g., Fig 2b, 2d)  show GKSN is much more stable than GMLP, especially with permutation invariance. This is a strong practical result. Do the authors have a clear intuition for why the KAN-based superposition architecture provides this added stability compared to a standard MLP?

---

> ### Author Response · Authors · 2025-11-24
> **Clarifications**
>
> We thank the reviewer for the feedback, and we address the remaining concerns.
>
>
> >The paper's key theoretical improvement reduces complexity from a naive $O(m^4)$ to $O(m^2n^2)$.[...]
>
> Please note that the theorem we provide has the following consequences: 1) the exact representation as mentioned reduces the complexity from $O(m^4)$ to $O(m^2n^2)$; 2) in the practical network implementation we additionally have that the number of input reduces from $O(m^2)$ to $O(nm)$, therefore the overall complexity becomes linear in the number of input, making it highly competitive; 3) the result extends also to the EMLP; 4) we further highligy that, since this is a theoretical results, it is not clear that it is possible to futher reduce the complexity of for an exact representation.
>
> >The authors motivate GKSN using the exact representation property of KAT [...].
>
> The KAT requires only continuity for functions (with a compact support). ReLU is in $C^0$, and can therefore be considered a continuous function, which is guaranteed to be learnable by KAT but used to bypass the redundant training process of learning it from scratch. Since ReLU is not in $C^1$ and it is in general not true that any equivariant function can be derived as the gradient of invariant functions, we can not use it for the alternative equivariant representation based on the gradient and we are only allowed to use it for the equivariant GKSN architecture which is not realized as the gradient.
>
>
> >Could the authors comment on the expected performance of a naive KAN (without the invariant-feature preprocessing) on these tasks? This baseline seems critical to understanding whether the performance gains stem from the KAN architecture itself or from the new geometric representation.
>
> Thank you for the comment. Using the KAN directly for the dataset would practically fail, since whenever we rotate the atomic system, the network is not able to know that the prediction should be the same. Hence, the generalization requirements for KAN are extremely amplified compared to our geometric approach, explicitly incorporating the necessary bias.
> In other words, our representation is always an invariant function for a specific group, while this is not true for the generic KAT representation. On the other hand, an invariant function is also representable with KAT.
>
> We extend on this point more technically: In order to address the effect of a group action with respect to a specific group G, there are three possible strategies: a) data augmentation; b) canonicalization, c) invariance of the network architecture, d) frame average. To augment the data, one artificially adds samples to which the group action is applied, for example by rotating the system. This has as consequences that 1) we need more data for training and 2) the invariance is always approximated; With canonicalization, we can try to guess the correct orientation (or predict, for example, by eigen decomposition). We apply the transformation, and then we can use a non-invariant model. The rotation can be learned with an additional model or using a Principal Component Analysis (PCA) on the whole structure of the molecule. If the canonicalization is correct, the model shows invariance, but since the canonicalization is not unique in the general case, the invariance is not guaranteed. The last approach is what we explore with our work. This approach is more principled, where we design architectures that “bakes-in” the invariance. The fundamental question we wanted to address is what the KAT representation (which is exact) looks like when we want to represent invariant functions to a class of group actions. We cover with our work some of the most important cases, i.e., the rotation group O(n) and the O(1,n-1) cases, and we cover invariant polynomial functions to the GL(n) group actions. An extension of this canonicalization is the frame average (FA) framework, where multiple group actions are applied to the input and their results are combined; for continuous group action, the transformation is only approximated.
>
> > The empirical results (e.g., Fig 2b, 2d) show GKSN is much more stable than GMLP, especially with permutation invariance. This is a strong practical result. Do the authors have a clear intuition for why the KAN-based superposition architecture provides this added stability compared to a standard MLP?
>
> This is an interesting question. We have not investigated the reason. There is no clear consensus on the reason why the GKSN model is more stable than the GMLP.
> KAT is used in the analysis of differentiable equations because it provides a way to separate variables. The intuitive answer would then be that the dataset exhibits features that tend to factorize on the input variables; therefore, GKSN provides a better bias than GMLP in this application.
>
> We hope we have addressed the reviewer’s concerns and, therefore, would update the score accordingly.

---

> > ### Comment · Reviewer_wwYp · 2025-11-26
> >
> > Thanks for the response. I maintain the score after checking the rebuttal and reviews from other reviewers. The current score reflects the merits of the current manuscript, and a score of 5 is actually more accurate.

---

### Meta-Review · Area_Chair_F5fc · 2026-01-04

**Summary:**

This is a borderline paper. It has two reviews below the threshold (4) and two reviews slightly above (6). One of the reviews above the threshold mentioned that they would give the paper a 5, and one of the reviews below said that they were not comfortable reviewing the mathematical formulation of the problem.

This is why I decided to check the mathematical formulation myself. The general approach is correct: they propose to use invariant theory to parameterize invariant KANs. However, the mathematical formulation related to the GL action is a bit sloppy.
In the background section, page 3 lines 131-138, the GL action is not mentioned. But in the paragraph afterwards, the fundamental theorem is stated for the GL action on pairs of objects $V^p \oplus V^{*q}$, which are the contractions between vectors and co-vectors. But then, Theorem 4.3 aims to apply the fundamental theorem for GL for a function that is $f:X^m\to\mathbb R$ (so there are no vectors and co-vectors involved, only vectors apparently). Maybe there is a natural identification they are using, but they do not acknowledge this important point. I checked in the appendix to see if I was missing it, and I found that lines 848 and 849 aim to identify the space with the dual somehow, but it uses a matrix A that is not defined.

I assess that this paper would benefit from a revision to make sure that the mathematical formulation is correct and clear. A simple example showing how to apply Theorem 4.3 in a toy example would be useful.

**Reviewer Concerns:**

Most of the concerns had to do with the experimental evaluation of the models. The authors addressed those concerns by providing more experiments. Minor concerns had to do with the computational complexity and were addressed correctly. Some of the reviewers could not follow the mathematical formulations.

**Reviewer Scores:**

I am not sure how the reviewers would have changed their scores.

---

### Decision · Program_Chairs · 2026-01-26

Reject